# Generalization properties of feed-forward neural networks trained on Lorenz systems

Sebastian Scher[1] and Gabriele Messori[1,2]

[1]Department of Meteorology and Bolin Centre for Climate Research, Stockholm University, Stockholm, Sweden
[2]Department of Earth Sciences, Uppsala University, Uppsala, Sweden

**Correspondence:** Sebastian Scher sebastian.scher@misu.su.se

**Abstract.** Neural networks are able to approximate chaotic dynamical systems when provided with training data that covers all relevant regions of the system's phase-space. However, many practical applications diverge from this idealised scenario. Here, we investigate the ability of feed-forward neural networks to: 1) learn the behaviour of dynamical systems from incomplete training data, and 2) learn the influence of an external forcing on the dynamics. Climate science is a real world example where these questions may be relevant: it is concerned with a non-stationary chaotic system, subject to external forcing and whose behaviour is known only through comparatively short data series. Our analysis is performed on the Lorenz63 and Lorenz95 models. We show that for the Lorenz63 system, neural networks trained on data covering only part of the system's phase-space struggle to make skillful short-term forecasts in the regions excluded from the training. Additionally, when making long series of consecutive forecasts, the networks struggle to reproduce trajectories exploring regions beyond those seen in the training data, except for cases where only small parts are left out during training. We find this is due to the neural network learning a localised mapping for each region of phase-space in the training data rather than a global mapping. This manifests itself in that parts of the networks learn only particular parts of the phase-space. In contrast, for the Lorenz95 system the networks succeed in generalising to new parts of phase-space not seen in the training data. We also find that the networks are able to learn the influence of an external forcing, but only when given relatively large ranges of the forcing in the training. These results point to potential limitations of feed-forward neural networks in generalizing a system's behaviour given limited initial information. Much attention must therefore be given to designing appropriate train-test splits for real-world applications.

## 1 Introduction

### 1.1 Neural networks for weather and climate applications

Neural networks are a series of interconnected – potentially nonlinear – functions, whose mutual relations are "learned" by the network by training on data. One of their many applications is forecasting the time-evolution of dynamical systems. In this context, the neural networks are trained on long timeseries issued from the dynamical system of interest, and can then

in principle be used to forecast the system's evolution from new initial conditions. Examples of applications include classical physical systems like the double pendulum (Bakker et al., 2000), and the widely studied Lorenz toy-models of the atmosphere (e.g. Vlachas et al. (2018); Dueben and Bauer (2018)).

In recent years, neural networks have enjoyed growing attention in climate science. Applications include parameterization schemes in numerical weather prediction and climate models (Krasnopolsky and Fox-Rabinovitz, 2006; Krasnopolsky et al., 2013; Rasp et al., 2018), post-processing of numerical weather forecasts (Rasp and Lerch, 2018), empirical error correction Watson (2019), predicting weather forecast uncertainty (Scher and Messori, 2018) and doing actual weather forecasts and climate model emulations in simplified realities (Dueben and Bauer, 2018; Scher, 2018; Scher and Messori, 2019), as well as doing actual weather forecasts (Weyn et al., 2019). These increasingly widespread practical applications warrant a more systematic evaluation of the possibilities and limitations of neural networks for the simulation of complex dynamical systems.

In this paper, we focus specifically on the widely used feed-forward neural networks and address two open questions related to their use for approximating the dynamics of chaotic systems:

1) Can neural networks infer system behaviour in regions of the phase-space not included in the training dataset?

2) Can neural networks "learn" the influence of an external forcing driving slow changes in the system they are trained on?

We adopt an empirical approach: we generate long time-series with numerical models, and then perform experiments with neural networks on this data. We specifically use the Lorenz63 (Lorenz, 1963) and Lorenz95 (Lorenz, 1996) models. These (and other variants of the Lorenz95 system) are widely used as toy-models for studying atmosphere-like systems, also in the context of machine learning (e.g. Vlachas et al. (2018); Watson (2019); Lu et al. (2018); Chattopadhyay et al. (2019)) and parameter optimization (e.g. Schevenhoven and Selten (2017)).

Both the questions we raise are of direct relevance to climate applications. Our knowledge of the high-frequency evolution of the climate system issues from comparatively short timeseries, which only explore a small subset of the possible states of the system. This is particularly true for the ocean, which has much longer characteristic timescales than the atmosphere, and for applications to paleoclimatic variability. Moreover, the accelerating anthropogenic forcing will likely lead to significant changes in the climate's future evolution. The two points we raise are therefore crucial in the context of using neural networks for weather forecasting and for emulating climate models. They could be reformulated in more practical terms as: do neural networks have the potential to reproduce unprecedented states of the climate system? Similarly, could they learn the influence of unprecedented greenhouse-gas concentrations on the dynamics of the climate system, given a past record of the system subjected to varying greenhouse-gas levels?

## 1.2 Related work on generalization properties of neural networks

The question of generalization is a central aspect in machine-learning, and is a well studied topic for neural networks (e.g. Hochreiter and Schmidhuber (1995); Hardt et al. (2015); Zhang et al. (2016)). One of the remarkable properties of deep neural networks is that, contrary to statistical learning theory, in many cases they generalise better when having more free parameters. The recent success of deep neural networks in a variety of applications and their empirically demonstrated generalisation abilities have stimulated investigations into the underlying mechanisms. For example, (Wu et al., 2017) argued that the reason

for their good generalization properties are the landscape characteristics of the loss function. Novak et al. (2018) argue that generalization is favoured by high robustness to input perturbations of the trained networks in the vicinity of their training manifold, despite their large numbers of parameters. Another well-studied aspect is machine learning under covariate-shift – the situation where the probability distribution of training and test data is not the same (e.g. (Sugiyama and Kawanabe, 2012)). This amounts to a special class of non-stationarity problems, and is partly related to our question 2 (learning external forcings).

The bulk of the literature on the above topics has focussed on image recognition and related fields, and the extent to which these results may apply to dynamical systems is unclear. To the authors' knowledge, the generalization properties of neural networks applied to dynamical systems, and specifically to Lorenz systems, are yet to be studied in detail.

## 2 Emerging Challenges in Neural Networks for Dynamical Systems

Question 1) we framed above, relates to whether the network learns a "global" function mapping the state vector $x$ from one timestep to the next:

$$f\left(x\right):x_t \mapsto x_{t+1} \tag{1}$$

or whether it learns $N$ individual functions for $N$ different regions of the phase-space:

$$f\left(x\right)=\begin{cases}f_1\left(x\right) & x \in region_1 \\ f_2\left(x\right) & x \in region_2 \\ \vdots & \vdots \\ f_N\left(x\right) & x \in region_N\end{cases} \tag{2}$$

Even though mathematically equivalent, the latter would imply that different parts of the network are responsible for different regions of the phase-space. For some applications this may be irrelevant, as long as the network forecasts work. However, it has major implications for how the network generalizes to regions of the phase-space that are not covered in the training data.

Neural networks can tend to overfit – meaning they work very well on the training data, but do not generalize and therefore do not work on new data. Therefore, they are usually tested on data not used for the training. Given a dataset, it is not trivial to decide how to split the data into a training and test set. For data without auto-correlation, a random split on a sample-by-sample basis may be suitable. For autocorrelated time-series, it is common to split the data into continuous blocks (e.g. using the first 80% of a timeseries for training, and the last 20% for evaluation). In a real-world application to the atmosphere, one could train the network on the first years of available observations, and then test on the remaining available years (e.g. Rasp and Lerch (2018); Scher and Messori (2018)). For the Lorenz models, the train-test splits are typically designed such that samples in the test set are not contained in the training set, but at the same time ensure that both the training and the test sets cover all regions of the phase-space with some reasonable density. That is, no large contiguous regions of the phase-space are left out of either set of data. (e.g. Pasini and Pelino (2005); Vlachas et al. (2018)).

Here, we consider the opposite situation, namely a scenario where the training data covers only part of the system's phase-space. We know from the definition of the Lorenz63 and Lorenz95 models that the underlying equations are invariant across the phase-space. If the network can truly learn the system's dynamics, and thus successfully approximates the underlying equations, then it should be able to provide useful information concerning the system's behaviour in those regions of the phase-space not included in the training data. More generally, for a long series of successive forecasts the network should thus be able to reconstruct the full attractor. However, should the network instead learn a set of functions each applicable locally, then one would expect the network to fail in regions not explored during the training. In a climate science context, this would for example be relevant for the ocean. The latter's long characteristics timescales imply that observational datasets may cover only part of the phase-space. It is also relevant in forecasting extreme events in the atmosphere.

Question 2) relates to how well a network can learn the influence of a slowly varying variable (the "forcing" in a general sense) on the evolution of the fast-varying variables (the system state). The influence of the slowly-varying forcing on the short-term dynamics is potentially very small compared to the typical variability of the systems, making the task of learning simultaneously the dynamics and the influence of the external forcing challenging, even when the forcing is provided as additional input to the network.

## 3   Methods

### 3.1   The Lorenz63 and Lorenz95 models

The Lorenz63 model (Lorenz, 1963) is a 3-variable system defined by the following ordinary differential equations:

$$
\begin{aligned}
\dot{x} &= \sigma\left(y - x\right) \\
\dot{y} &= x\left(\rho - z\right) - y \\
\dot{z} &= xy - \beta z
\end{aligned}
\tag{3}
$$

We use $\sigma = 10$, $\beta = 8/3$ and $\rho = 28$, the standard parameter combination with which the system – despite its simplicity – generates chaotic behavior (the characteristic "butterfly" shape, see Fig. 1a). We integrate the system with a timestep of $t = 0.01$ with the LSODA solver from ODEPACK (Hindmarsh, 1983) as provided by scipy (Jones et al., 2001). While the Lorenz63 model is a very rough approximation of atmosphere-like dynamics, the fact that it only has 3 variables allows to easily visualize the complete phase-space and define regions that can be excluded from the training data. This makes it ideally suited to tackle the first question we pose (generalization to unseen phase-space regions).

The Lorenz95 model ((Lorenz, 1996), also often referred to as the Lorenz96 model) is a 1-d model that approximates the atmosphere as a series of $N$ gridpoints wrapped around a circular domain:

$$
\dot{x}_i = \left(x_{i+1} - x_{i-2}\right) x_{i-1} - x_i + F
\tag{4}
$$

with $i = 1...N$ and $(x_{N+1} = x_1)$. Here we choose $N = 40$. $F$ is a forcing term. With $F = 4$ the system shows periodic behaviour; with increasing F the behaviour becomes increasingly chaotic, and with $F = 16$ it is highly turbulent. An example of a Lorenz95 model integration is shown in the left panel of Fig. B1 d. Note that there is also a second model often referred to as the Lorenz95 or Lorenz96 model, which uses a second (and sometimes a third) dimension. This model is not considered here. Like the Lorenz63 model, we integrate the system with the LSODA solver from ODEPACK.

## 3.2 Neural Network for Lorenz63

For the Lorenz63 model we use fully-connected networks with ReLu activation functions in the hidden layers and a linear output layer. The main configuration used in this study was determined via a tuning procedure (Appendix A). It consists of 2 hidden layers with 128 neurons each. The network takes as input all 3 Lorenz63 variables, and outputs all 3 variables one timestep later. The training is done with the adam optimizer (Kingma and Ba, 2015). Overfitting is controlled via an early-stopping rule. The training is stopped when the skill on a validation dataset (last 10% of the training set) has not increased for 4 training epochs, with a maximum of 100 epochs. For the forcing experiments, we additionally use a second architecture, where the network has 4 input parameters (the 3 Lorenz63 variables and the parameter $\sigma$, see Eq. 3), and the same 3 output variables as the standard setup. No regularization techniques are used. Part of our experiments are repeated with the same architecture, but trained on forecasting the tendency (difference between the following and current states) rather than the following state directly.

## 3.3 Neural Network for Lorenz95

For the Lorenz95 model, we use a convolutional network that works on the periodic domain. Convolutional networks have already successfully been used on gridded data from simplified general circulation models in Scher (2018) and Scher and Messori (2019). The configuration used here was tuned with an exhaustive gridsearch over different network configurations. The tuning procedure is described in Appendix B. We tuned the network for forecasting 1, 10 and 100 timesteps, where each timestep corresponds to 0.01 time units of the Lorenz95 model. The network trained for 10 timestep-forecasts (a 2-layer convolution network with a kernel-size of 5, see Appendix B) worked best for virtually all lead-times (see Fig. B1), and we use this architecture in our analysis. For the forcing experiments, the parameter $F$ at each timestep was expanded to the number of gridpoints of the Lorenz95 model and added as an additional input channel to the network. As for the Lorenz63 model, the network directly forecasts the next state of the system. Overfitting is controlled via an early-stopping rule. The training is stopped when the skill on a validation dataset has not increased for 4 training epochs, with a maximum of 30 epochs. No regularization techniques are used.

## 3.4 Evaluating the reconstruction of the Lorenz63 attractor

In most of our experiments, the neural networks are trained by minimizing errors of single-step (and thus short-term) forecasts. Therefore, they may not always reproduce a stable system when making a long series of consecutive forecasts – a known issue

when applying neural networks to chaotic systems (e.g. Bakker et al. (2000)). For the Lorenz63, the trained network often made very good short-term forecasts, but when attempting to produce long series of iterative forecasts (which, in the context of climate science, would be analogous to producing a "climate run" from successive meteorological forecasts), the system collapsed into a fixed point. Since the training of our network is computationally inexpensive, we use a brute force method to find a network that yields both skillful short-term forecasts and a realistic long-term system evolution. We train 10 networks, and then select the network that best reproduces the attractor when started from a random point in the training dataset. This is evaluated via comparing the reconstructed attractor to the training data using:

$$rmse\,(\rho) = \sqrt{\left(\rho_{i,j,k,model} - \rho_{i,j,k,network}\right)^2} \tag{5}$$

where $\rho_{i,j,k}$ is the density of discrete data points in the gridbox $i,j,k$. We will hereafter term this the "density-selection" approach. The gridboxes have size $0.3 \times 0.3 \times 0.3$ on the normalized domain (normalization based on the training set, the output of the networks is always in the normalized domain).

This approach is somewhat problematic when training the network on specific regions of the phase-space. In principle, we could apply exactly the same procedure to compare the densities of the reconstructed attractor and of the training data. However, for incomplete training data – for example, only one wing of the butterfly – then a perfect reconstruction of the full attractor would fail this test, since the training data includes no information beyond the one wing. If the neural network were to learn a "wrong" attractor, namely one that only covers regions close to the wing included in the training, this network would pass the test and be selected, even though it clearly has undesirable characteristics. An alternative approach is to compare the reconstructed attractor with the full attractor. This solves the aforementioned problems, yet is flawed in terms of information availability at time of training. In a real world setting, we would not know what the full attractor of a complex system – for example our atmosphere – looks like. Nonetheless, in our idealised setting this approach allows to verify whether the network learns regional or global dynamics. We will hereafter term it the "density-full approach".

## 4    Reconstructing Lorenz systems using only part of the phase-space

### 4.1    Lorenz63

#### 4.1.1    Training the networks

We first verify that our networks can successfully reproduce the Lorenz63 attractor given training data from across the system's phase-space. We train 10 networks on a long Lorenz63 simulation (1e6 timesteps) meant to explore all regions of the butterfly, and make forecasts 0.01 time-units ahead. The networks are then initialized with a random state out of the test dataset, and we make 1e6 consecutive forecasts (via feeding the forecast back into the input). Figure 1 a,b shows the training data and the attractor reconstructed by the neural network. The network attractor reproduces the typical "butterfly" shape and, most importantly, it neither drifts into a periodic orbit nor collapses into a fixed point. Its main deficiency is that the inner regions of

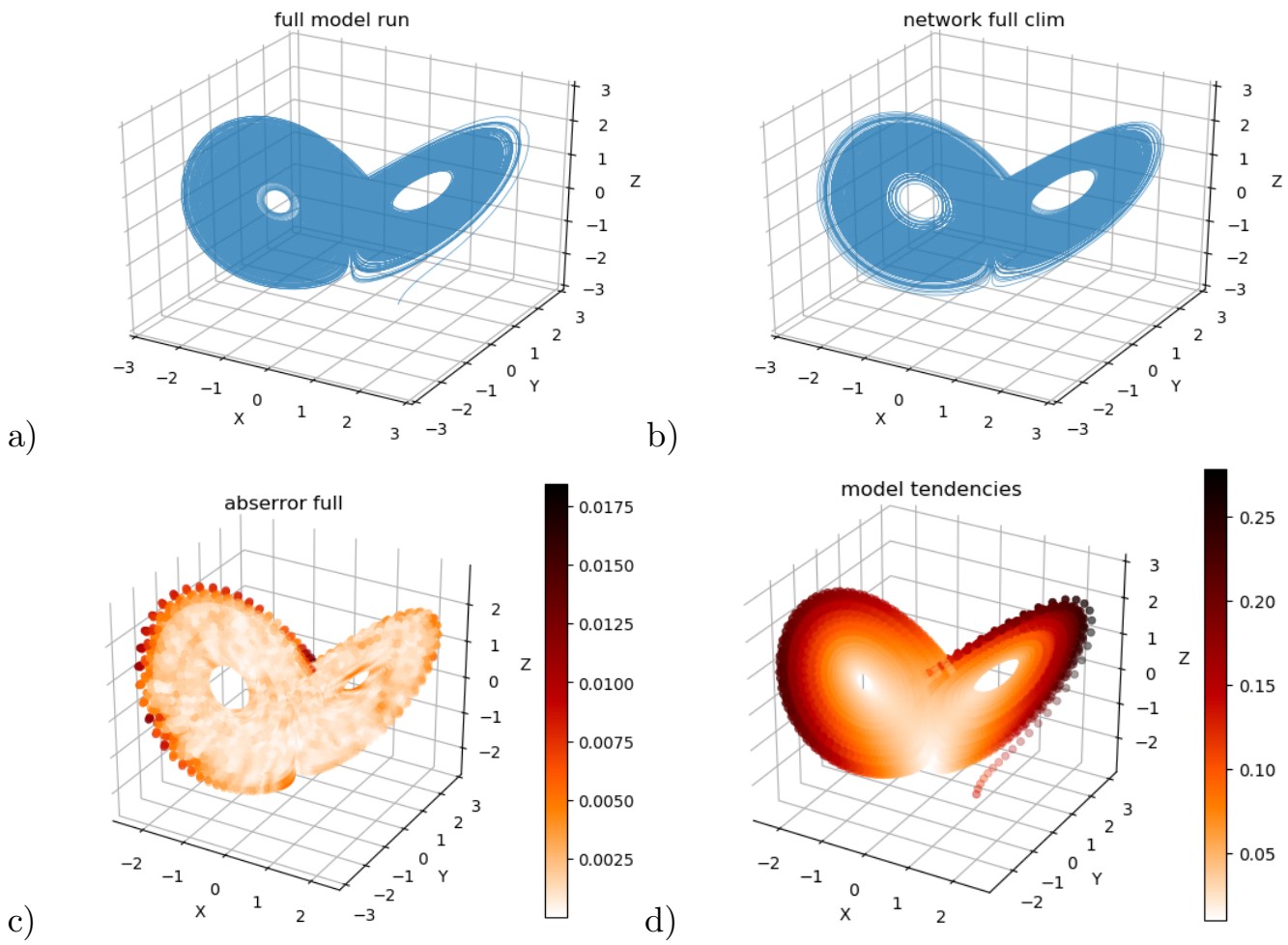

**Figure 1.** a) A long integration of the Lorenz63 model. b) Timeseries produced with a neural network optimized on short-term forecast error, initialized from a random initial state not used in the training. c) Short-term forecast errors of the neural network initialised at a large number of points not used for training. d) tendencies for 1 timestep ($t_{i+1} - ti$). Note different colorscales in c) and d).

the wings are slightly underpopulated. Figure 1 c) shows the mean absolute error (MAE) of 1-step network forecasts initialized at every point in the test set. The forecasts typically display small errors (<0.03). The highest errors occur in the edges of the wings, where recurrences are rare and the intrinsic predictability of the system is low (Faranda et al., 2017). To put forecast errors throughout the paper in context, panel d) shows the tendency (change over 1 timestep) of the model in different phase-space regions.

We next consider the question of training on incomplete data. We take a somewhat drastic approach and we select data that explores only limited regions of the phase-space. This selection is done via "cutting out" contiguous regions of the phase-space. Since the training is done on data pairs (timesteps $t_i$ and $t_{i+1}$), the points at the locations where the trajectories are truncated

are removed from the training data to avoid artificial "jumps" towards the next included point (this is necessary because we removed parts of the model's trajectory). First, we investigate whether neural networks trained on different phase-space regions are able to make short-term forecasts in other parts of the attractor. Then, we assess whether it may be possible to reconstruct the full attractor with these networks.

### 4.1.2 Short-term forecasting

Figure 2 shows the short-term forecast error for a network trained only on the left wing (a,d), only on the right wing (b,e) and on a butterfly with a truncated right wing tip (c, f). In the wing where training data was present, the forecast error is very similar to the error of the network trained on the full attractor (Fig. 1 c). In the wing that was excluded during training, the forecast error is much higher. It is in fact so high (mean absolute error on the order of 0.7) that the forecasts have little to do with the real system. Closer examination reveals that when initialized in the "missing" wing, the forecasts point back towards the "training" wing (see Sect 4.1.3). When excluding only the tip of the right wing, the network manages to make somewhat reasonable forecasts in the "missing" region, and does not systematically point back to the region seen in the training. Nonetheless, the forecast errors in the "missing wingtip" are roughly an order of magnitude higher than in the regions included in the training (Fig. 2 c,f). These findings suggest that the network does not learn a global mapping, but a localized one which fails in previously unexplored regions. The results are similar when using networks that forecast the tendency only instead of the following state (Fig. C1). The main difference is that when training on only one wing, the error in the other wing is roughly halved relative to Fig. 2, albeit still orders of magnitude higher compared to training on the whole attractor. When initializing forecasts in the left-out wing, the trajectories are unstable and drift outside of the training domain (not shown). In this respect, the architecture that forecasts tendency is doing even worse than the architecture forecasting the state. The simplicity of the system allows us to examine the above results further by looking at the activation of the individual neurons in the network. For this, we inspect a network that was trained on the whole attractor (and provides good forecasts on the whole attractor).

Figure 3a,b shows the distribution of activations (i.e. output) of the hidden neurons for the network trained on the whole attractor, when fed with input from the left wing only (green) and from the right wing only (orange). Shown are the 20 neurons with the largest absolute differences in the standard deviation of activations in the two wings. The distribution of activations for all neurons (without specific ordering) is shown in Fig. C2 in the appendix. Some neurons have very similar activations in both wings, whereas the distributions of other neurons change significantly. In both wings, some of the neurons have very little spread in activation, meaning that their output is relatively independent of the exact location within the wing. However, these "low-variance" neurons are not the same in the two wings. We hypothesize that they correspond to a localized mapping that the network learned for the other wing. This would mean that the neurons learned to correctly map the system in one wing – and are thus active and contributing to the forecasts in that wing – but they are inactive in the other wing (i.e. do not contribute to the forecasts). To test this, for each layer we identify the $n$ neurons with least spread in activation (defined here as standard deviation of the activation) for all points on each wing. We then create modified networks by fixing the output of each of these neurons in turn at their mean activation level for the relevant wing. Note that there are also some "dead" neurons, which always have zero activity in both wings. These we ignore. With this modified network, we make forecasts on the whole

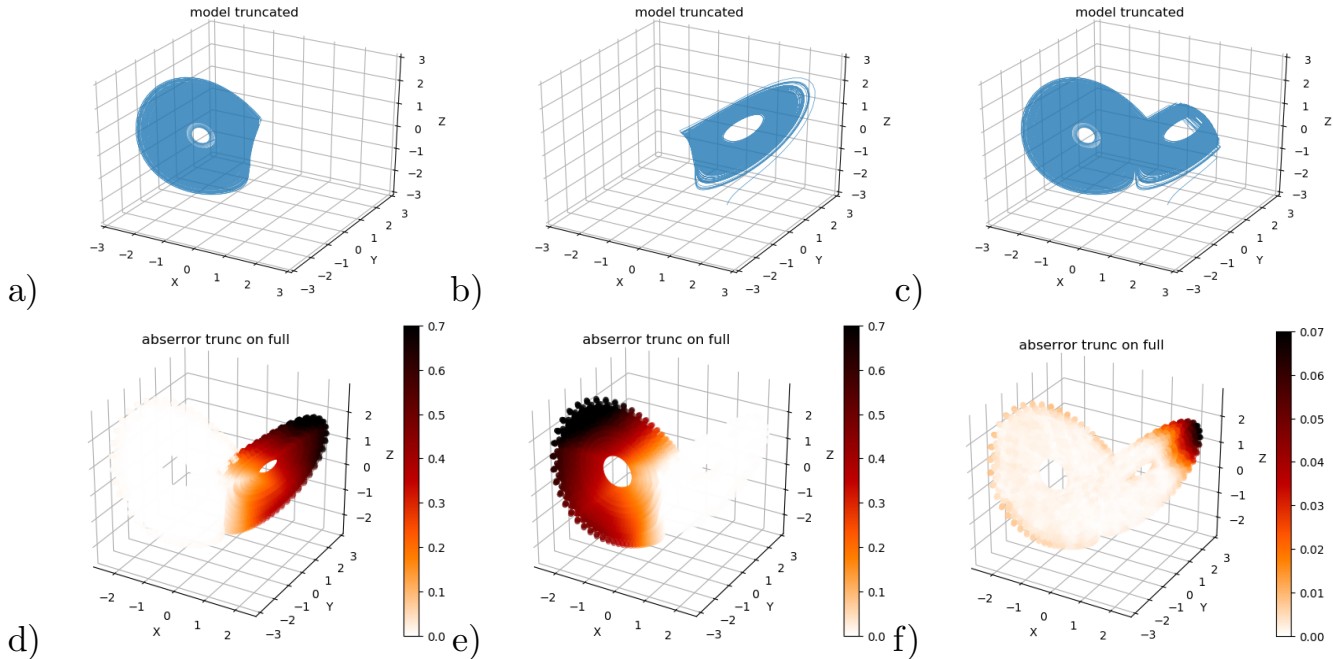

**Figure 2.** Truncated sets of Lorenz63 training data (a-c) and short-term forecast error (MAE) of neural networks trained on these sets (d-f). Note the different colorscales in (d-f).

attractor. The result for $n = 20$ is shown in Fig. 3 c,f. The effect of fixing the activations of the neurons that have low spread in the left wing is that the forecast error in part of the right wing increases sharply, whereas the error in the left wing is nearly unchanged. The same is seen for forecasts in the left wing when fixing the activations of the neurons that have low spread in the right wing. The structure of the errors is very similar to that of the networks trained only on one wing (Fig. 2). Panels 3 e,f give a more systematic overview. They show the forecast errors of the modified networks on the left wing (green) and the right wing (orange), when fixing the 1,2,...100 neurons the have lowest variance in the left and right wings, respectively. When fixing the left wing "low-variance" neurons, the error in the right wing increases with even a single deactivated neuron, and rises monotonically with every additional deactivation (Fig. 3 e). In the left wing, on the other hand, the error stays very close to the error of the unmodified network, and only starts to increase beyond 20 deactivated neurons. Corresponding results are found when fixing low-activity neurons in the right wing (Fig. 3 f).

The above suggests that these roughly 20 neurons correspond to the localized mapping part of the network we had speculated about earlier, and deactivating them forces the network to fall back to its global mapping, which we have seen is poor. This test was repeated for different network architectures (different number of hidden layers, and different hidden layer sizes). In all we tested 20 different architectures (smallest: 1 hidden layer with 8 neurons, largest: 8 hidden layers with 128 neurons each). The result for eight of these architectures (ranging from shallow networks with narrow layers to deep networks with wide layers) are shown in Fig. 4. The behaviour is very similar to that seen in Fig. 3, except that in some cases the error does not

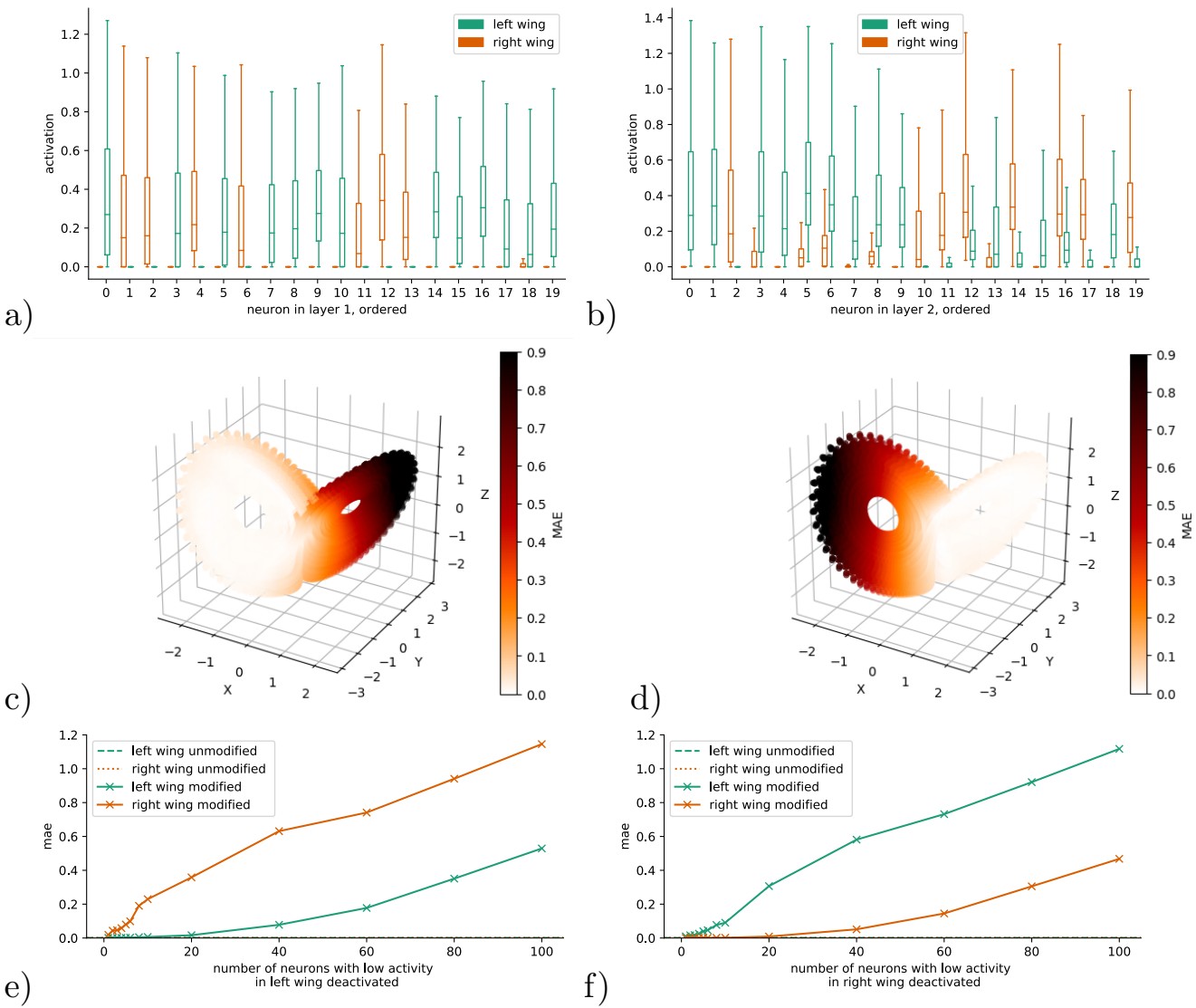

**Figure 3.** Boxplots showing the distribution of neuron activations per neuron for hidden layer 1 (a) and hidden layer 2 (b), split by wing (color in plot). Short-term forecast errors (MAE) for the networks in which in each layer the activation level of the 20 neurons with lowest variance are fixed at its mean value for the left (c) and right (d) wings. Short-term forecast errors of the network with 1–100 low-variance neurons per layer in the left (e) and right (f) wings deactivated, split up by wing (solid lines). The dashed lines show the forecast errors of the unmodified network.

grow monotonically with increasing number of deactivated neurons. The results for the additional architectures we tested were similar (not shown). The only exception is the deepest architecture with very narrow layers ( 8 layers with 8 neurons each), in which deactivating a single low-activity neuron per layer degrades forecasts in both wings (not shown).

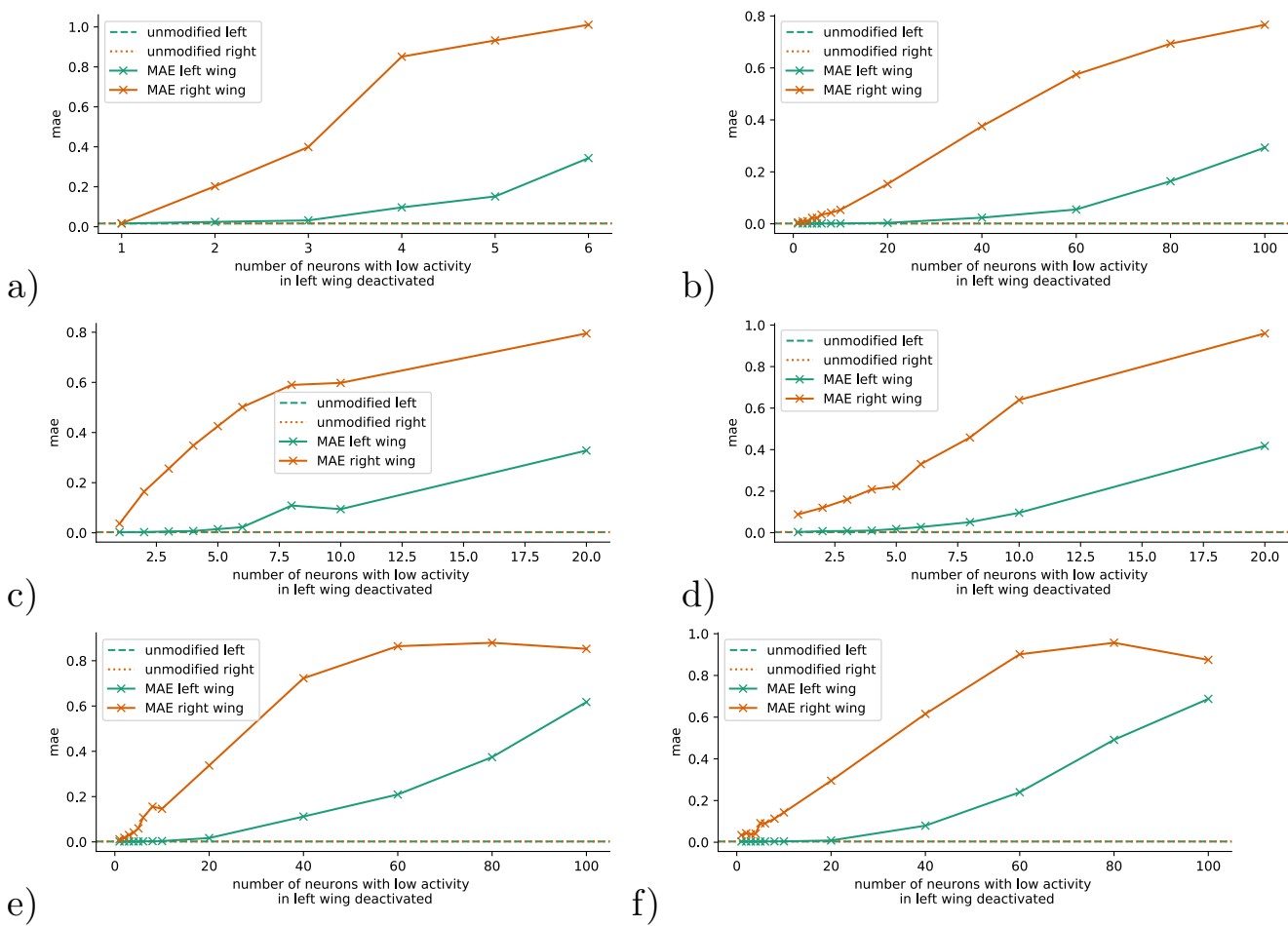

**Figure 4.** As Fig. 3e, but for different neural network architectures. a) shallow and narrow (1 hidden layer with 8 neurons); b) shallow and wide (1 hidden layer with 128 neurons); c) intermediate (2 hidden layers with 32 neurons each); d) deeper intermediate (4 hidden layers with 32 neurons each); e) deep and wide (4 layers with 128 neurons each); f) very deep and wide (8 layers with 128 neurons each).

### 4.1.3 Reconstructing the full attractor

We next attempt to use neural networks trained on incomplete data to reconstruct the full attractor. We already showed in Sect. 4.1.1 that this is possible when training on the whole attractor. When we remove only a small part of the attractor from the training data (the tip of the right wing, Fig.5 a), the networks are able to reproduce a reasonable attractor regardless of whether they are selected using the density-selection (Fig.5 b) or the density-full approach (Fig.5 c) – see Sect. 3.4. As could be inferred from the short-term forecasts, the neural networks are thus able to explore regions that are not visited by any of the trajectory segments in the training data. However, networks trained on single wings fail to reconstruct the full attractor (Fig.5 e, h), independent of the selection criterion used. These networks either failed the selection tests, or produced trajectories that populate only the wing used in the training. The networks also fail to explore the other wing when they are initialized from states within it. In this case, the trajectories immediately point back to the wing the network was trained on, and reach it after a couple of iterative forecasts (Fig.5 f, i), implying that the network reproduces a dynamics that populates only the wing that was included in the training.

### 4.2 Lorenz95

In the Lorenz95 system, which in our setup has a dimensionality of 40, it is harder to define reasonable regions of phase-space to be excluded from training than in the 3-dimensional Lorenz63 system. A logical step to tackle this problem would be to use a method like principal component analysis to reduce the dimensionality of the system before partitioning its phase-space. However, the leading principal component of the Lorenz95 system can only explain 8 % of the variance (not shown), meaning that it is not possible to reduce the system to a small number of principal components while still capturing most of its variance. A different approach is to look at Poincaré sections. These are 2-dimensional projections of the phase-space spanned by two variables, often used in the analysis of dynamical systems. While this approach seems intuitive, it is problematic in our context. If we define a region of the phase-space to leave out of the training (by defining a region spanned by 2 variables) we can cut out all states of the model run that fall within these regions. However, if there were identical states to these, but shifted one or more gridpoints, then these states would not be excluded. The symmetry of the system (which also translates to the symmetry in the circular convolutional network architecture used), implies that the network can forecast states excluded from the training data without learning any extrapolation, as long as (near-)identical but shifted states are seen while training. Indeed, due to the circular convolution, original and shifted states are equivalent for the network. Based on these considerations, we use another method to define Poincaré sections of the Lorenz95 system. We first transform the system states to spectral space with a Fast Fourier Transform (FFT). We then compute the amplitude of each wavenumber (absolute value of the complex wavenumber coefficients), thus removing all information about the position of the waves. We next find the pair of wavenumbers whose amplitudes have least correlation, and define a Poincaré section based on these.

Since the Lorenz95 model is very cheap to run, we can also – in analogy to the Lorenz63 experiment – define a phase-space region via setting a certain range for all 40 variables. Due to the low density of data points in such a high-dimensional space, this would exclude only very few points from our standard 1e5 timesteps run, and likewise, only few points in the test set

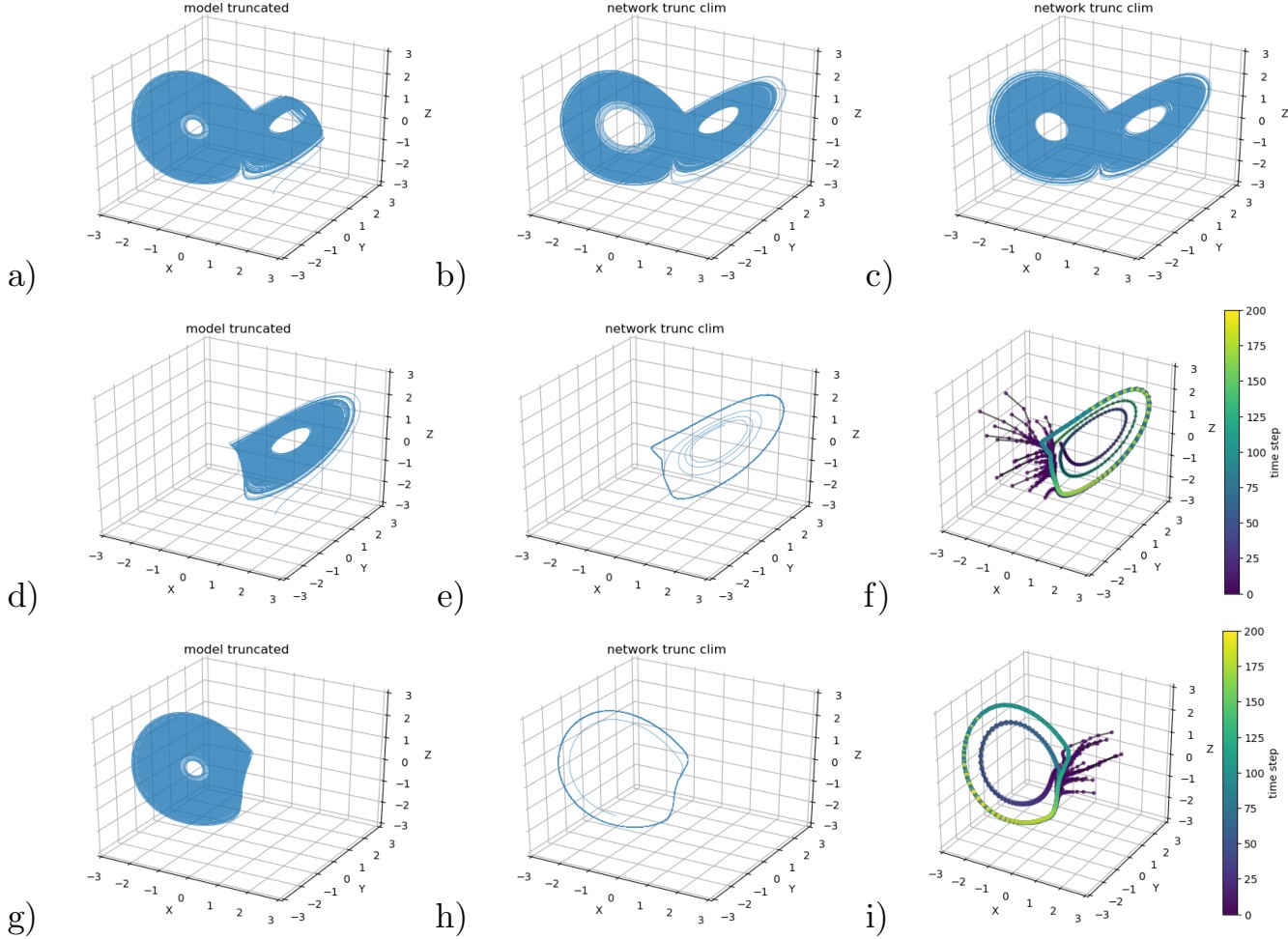

**Figure 5.** Reconstruction of the Lorenz63 system with neural networks trained on truncated data. a,d,g) Truncated sets of Lorenz63 training data. Reconstructed attractors with networks trained on (a) and selected using the density-full (b) and density-selection (c) approaches. e,h) Reconstructed attractors trained on (d,g) respectively, selected based on having no repeated points. f,i) trajectories initialized with random points from the region of the attractor that was left out in the training data in (d) and (g), respectively. The points in (f,i) indicate single forecast steps.

would lie in this region. Therefore, for this approach we generate an additional test-set. We run the model until we have 1e3 points that lie in the region cut our from training. Due due to the symmetry considerations mentioned above, we do this in the 20-dimensional space of absolute wave-number coefficients.

To implement the first method we "cut out" squares of the spectral Poincaré section, and train a network on the rest of the data. We then use the network to forecast the whole attractor on a test set, and compare it to the skill of the same network trained on the whole attractor (which has good forecast skill, see Appendix B and Fig B1). Each training is done 10 times, and the forecast errors averaged over these 10 realizations. The results are shown in Fig. 6a,b. The short-term forecast errors in the left-out region are indistinguishable from the errors in the other regions, meaning that the network does succeed in generalizing to regions not seen in training. This is also the case for other choices of left-out regions (not shown).

For the second method, we remove all training points that lie within than range $[0, 10]$ for every wavenumber. Again, the experiment is repeated 10 times. The result is shown in Fig. 6 c,d. Again, the short-term forecast errors in the region left out in the training are indistinguishable from errors in other regions. Also, the difference between the errors of the network trained on all points and those of the network trained on the truncated set is smaller than the difference between different training realizations (not shown). Finally, we test whether a long run of 1e4 consecutive NN forecasts explores the regions of phase-space left out from the training data. The runs were intialized from a random state of the test set not lying in the left-out region. For all 10 trained networks, the runs did explore the left-out region (not shown).

## 5  Learning external forcings of Lorenz Systems

### 5.1  Lorenz63

As external "forcing" scenario we consider a gradual linear increase of the $\sigma$ parameter (eq. 3). We train the network architecture using $\sigma$ as input (see section 3.2) on Lorenz63 runs with 1e5 timesteps, with linearly increasing $\sigma$ over the whole run. We perform 6 different runs, encompassing different $\sigma$-regimes regimes: two runs in a low (varying $\sigma$ from 7 to 8 and 6 to 9), two in an intermediate (10 to 11 and 9 to 12), and two in a high regime (12 to 13 and 11 to 14). The networks are then evaluated on a set of 10 Lorenz63 test runs (length 1e5 timesteps) with $\sigma$ fixed at 4, 5, 6, 7, 8, 8.5, 9, 10, 12 and 14, respectively. In addition to the main network, two references are used. Firstly, a network trained on the Lorenz63 run with varying $\sigma$, but not using $\sigma$ as input (termed "no input"). This network is then evaluated on the above fixed $\sigma$ runs. Secondly, for each run with fixed $\sigma$, an identical run but with different initial conditions is made. Then, a network not using $\sigma$ as input is trained on the latter run, and evaluated on the former run with the corresponding fixed $\sigma$ (termed "fixed $\sigma$"). The short-term forecast quality is assessed by initializing one-step forecasts from every state in the test runs, and computing the MAE. Each experiment is repeated 10 times, using the same training and test data, to capture potential influences of random components in the training.

The results are shown in fig. 7. Each panel represents a certain training range in the forcing (indicated by the grey area), and the lines show the MAE of one-step forecasts. The "fixed $\sigma$" networks (green lines) can be seen as a an upper baseline, as their skill is that obtained when training in the same forcing regime as used for evaluation. It is not expected that the main network (the one using $\sigma$ as input and trained on the run with linearly increasing $\sigma$) would do better than this reference. The "no

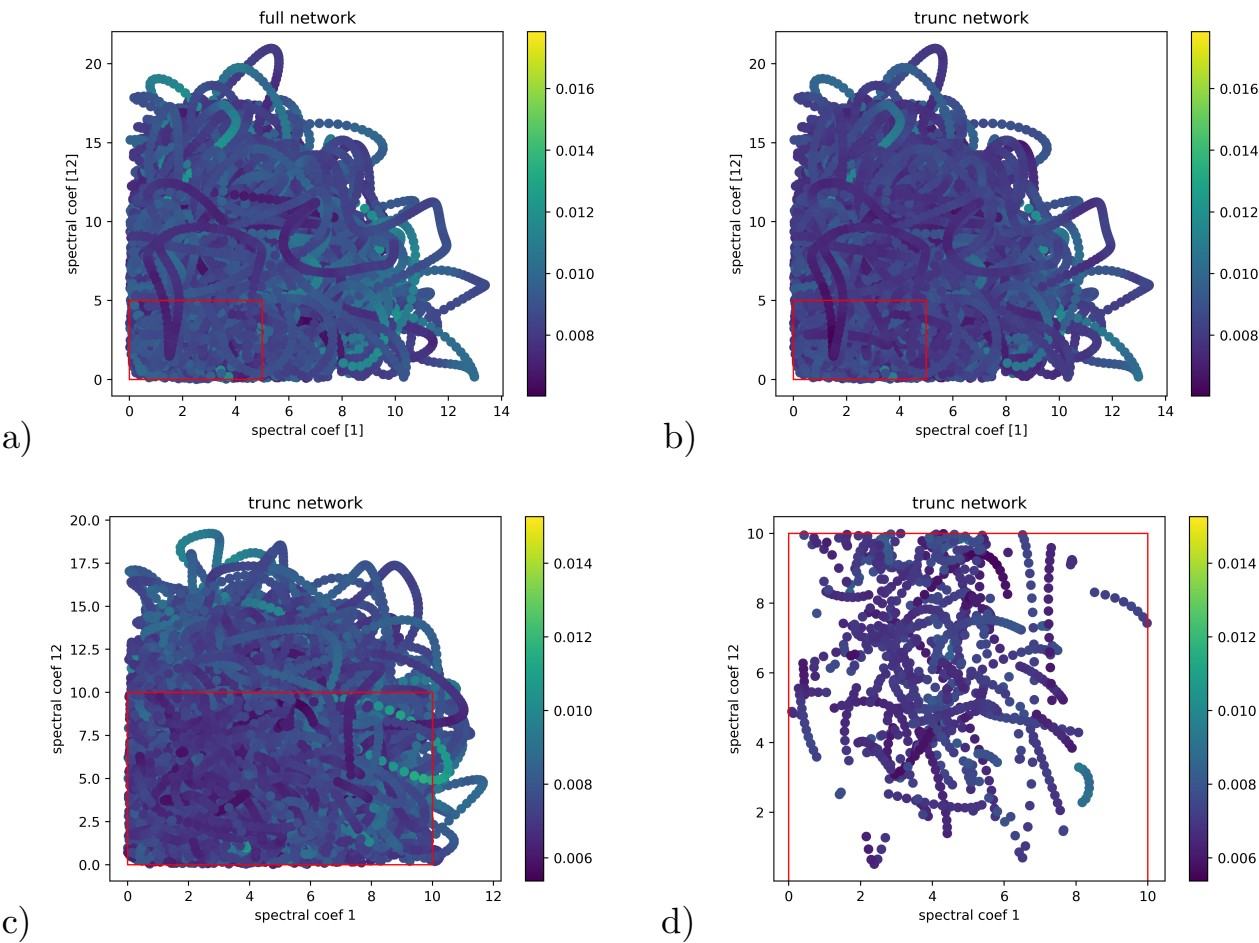

**Figure 6.** Networks trained only on part of the Lorenz95 phase-space. Short-term forecast errors of the network trained on full training set (a) and a truncated set selected on a Poincaré section in spectral space (b), projected onto said Poincaré section. The rectangle denotes the region of phase-space left out from training. c,d) Short term forecast errors of network trained on a truncated set selected on all 20 spectral components. c) shows all points in the test set, d) only the points that lie in the region cut out from training.

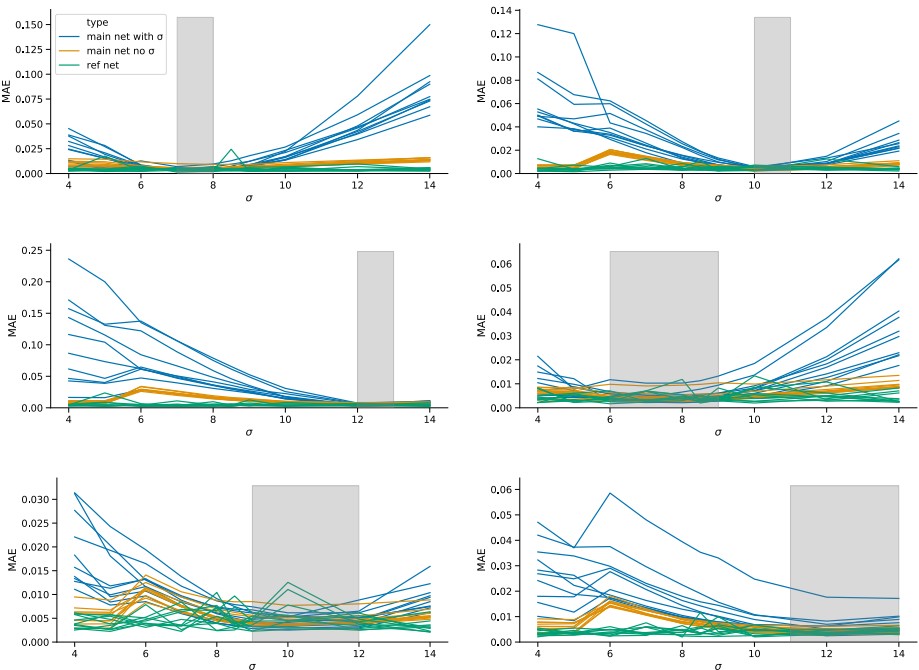

**Figure 7.** Short-term forecast errors for "forcing" experiments with Lorenz63. Networks are trained on a run with 1e5 timesteps with linearly increasing parameter $\sigma$ from the lower to the upper end of the grey shaded area in the panels, and then tested on runs with fixed $\sigma$. The blue lines show the errors for networks that include $\sigma$ as input. The yellow lines show the networks without $\sigma$ as input ("no input" networks in the text). The green lines show reference networks without $\sigma$ as input that are trained on runs with $\sigma$ fixed to the same value as the test runs ("fixed $\sigma$" networks in the text).

input" networks (yellow lines) can be used as a lower baseline, as this should be the skill that can be achieved without having any knowledge of the changing forcing. When trained on the narrow forcing regimes, the networks have trouble making good forecasts outside the training regime. The forecasts are indeed so poor that even the "no input" networks outperform them. In other words, the additional information provided by $\sigma$ actually leads to a deterioration in skill. This changes slightly when

5   training on broader regimes. Here, the forecast errors of the networks using $\sigma$ as input are similar to the "no input" networks, although in most cases they are still far from matching the "fixed $\sigma$" networks.

## 5.2   Lorenz95

We next consider a variable forcing scenario for the Lorenz95 system. The setup is analogous to the Lorenz63 forcing experiment, but here we change $F$ instead of $\sigma$. With $F = 4$, the system shows periodic behaviour; as $F$ increases, the system

10   becomes more and more turbulent. We consider two low (varying $F$ from 5 to 6 and from 4 to 7), two intermediate (8 to 9 and 7 to 11), and two high forcing regimes (12 to 13 and 11 to 14). The runs are evaluated for $F$ fixed at 4, 5, 6, 7, 8, 8.5, 9, 10, 12

and 14. In addition to evaluating short-term forecast performance as in the Lorenz63 forcing experiment, for the Lorenz95 we also asses the ability of the trained networks to reconstruct the "climate" (or attractor) of the model by making a 1e5 timesteps climate run with the network, and then computing the mean and standard deviation of the run (averaged over all gridpoints).

The results are shown in Fig. 8. Each row represents a specific training range in the forcing (indicated by the grey area). The left panels show the MAE of short-term forecasts, while the right panels show mean and standard deviation of the reconstructed climates, as well as mean and standard deviation of the Lorenz95 model. Each line represents on of the 10 runs made for each experiment. For the three experiments that are trained on narrow forcing regimes (5 to 6, 8 to 9 and 12 to 13), the main networks do not seem able to learn the influence of the forcing and extrapolate to new regimes. In all experiments, the main network has much higher short-term MAE than the "fixed $F$" networks. When trained on the low or middle regimes, the forecasts are even worse than those of the "no input" networks . As for the Lorenz63, the additional information provided by the forcing term therefore leads to a poorer performance of the network. This picture changes when training on broader forcing regimes (lower 3 rows in Fig. 8). Even though there is a large variation between the individual training realizations of the main network, both the ones trained on the high and on the intermediate forcing regimes outperform the "no input" networks. This implies that, given a wide enough forcing regime in the training, the network is able to learn – at least part of – the influence of the forcing on the dynamics, and extrapolate this influence to new forcing regimes

## 6   Discussion and conclusion

In this study, we explored how well feed-forward neural networks can 1) generalize the behaviour of a chaotic dynamical system to its full phase-space when trained only on part of said phase-space, and 2) learn the influence of a slow external forcing on a chaotic dynamical system. Both points are of direct relevance to the application of neural networks in climate science. The climate system is highly chaotic, our observational data likely includes only a small portion of the possible states of the system and we are subjecting the system to a slowly varying forcing by emitting large amounts of greenhouse gases. To address these points, we used two highly idealised representations of atmospheric processes, namely the Lorenz63 and the Lorenz95 models. We used feed-forward neural network architectures that are shown to work well on these systems when trained on the full phase-space and without external forcing.

For the first point we raise, we showed that networks trained on only part of the Lorenz63 attractor are largely unable to reproduce trajectories outside the regions they were trained on. When making short-term forecasts initialized from points in these unknown phase-space regions, the trajectories of the network forecasts point back towards the region included in the training. This makes the forecasts so poor as to be practically useless. Similar issues arise when running a large number of iterated forecasts, so as to reproduce a long trajectory of the system using the neural networks. Again, the network trajectories do not explore regions of the phase-space that were not included in the training. The only exceptions are cases where very small regions are excluded from the training data (and determining what is the limiting size of "very small" remains an open question). This implies that using neural networks for emulating climate models, as proposed in Scher (2018) and Scher and Messori (2019), may be more challenging than expected. The same goes for making forecasts of unprecedented weather or

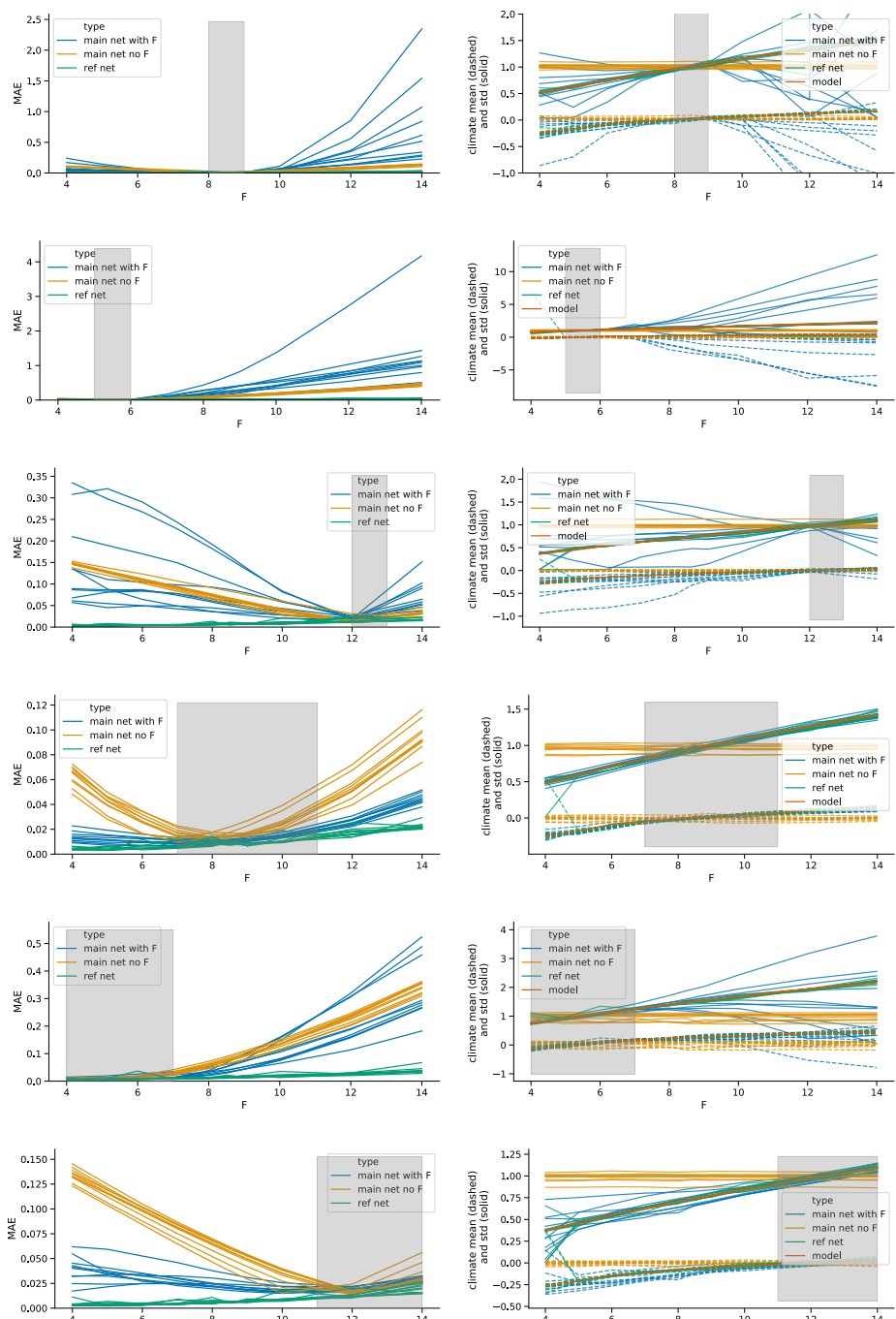

**Figure 8.** Short-term forecast errors and network attractor reconstructions for forcing experiments with Lorenz95. Networks are trained on a run with linearly increasing F from the lower to the upper end of the grey shaded area in the plots, and then tested on runs with fixed F. The blue lines show the network that include F as input. The yellow lines show the networks without F as input ("no input" networks in the text). The green lines show reference networks without F as input that are trained on runs with F fixed to the same value as the test runs ("no F" networks in the text). The left panels show short-term forecast error, the right panels show mean and standard-deviation of climate runs performed with the networks and additionally the mean and standard-deviation of Lorenz95 runs with F fixed to the reference values (orange lines)

climate events, or of events originating from unprecedented atmospheric or oceanic configurations. In contrast to the results for the Lorenz63 system, our experiments for the Lorenz95 system indicate that the networks can succesfully make forecasts in phase-space regions left out from training, and also explore these regions when making long simulations. In this respect, we have to note the difficulty of defining sensible regions of phase-space for the Lorenz95 system with its 40 dimensions. These difficulties would be even more severe for more realistic systems like numerical general circulation models. Still, this result is somewhat counter-intuitive, as one may naively consider the Lorenz95 system to be more complex than the Lorenz63 system. Therefore, our results indicate that using intuitive definitions of the complexity of a system to reason on the performance of feed-forward neural networks is problematic.

For the Lorenz63, we interpret our results as indicating that the neural networks do not learn to approximate the equations underlying the dynamics of the system — which would be akin to a "global mapping" — but rather develop a "regionalized view" of the system, whereby specific neurons contribute to the forecasts in specific regions of the phase-space. Thus, when parts of the phase-space are left out, the regionalized mapping fails to produce sensible estimates of the system's behaviour beyond the regions it has already seen. We confirmed this by inspecting the activations of individual neurons in the trained networks, and showed that parts of the network are responsible for specific regions of the phase-space. This is similar to findings in the context of image recognition and generation, where different parts of neural networks have been shown to represent different objects/concepts (Bau et al., 2019).

As a caveat, we note that our experiments, which remove a large contiguous region of phase-space from the training data, are more penalising than what may be expected in a typical climate simulation. It is likely that the regions of the phase-space explored by the climate system during the satellite era are more representative of the hypothetical climate attractor than a single wing of the butterfly is for the Lorenz63 system. Indeed, removing a wing is more akin to removing a season from a training set — for example asking a network to simulate a seasonal cycle without ever being trained on winter data — than having a training set which does not include some rare extreme events — which presumably live in sparsely populated regions of the phase-space which need not be contiguous.

An additional challenge in this context that became obvious during the design of our experiments is the choice of criteria to judge successful attractor reconstruction after training. As discussed in the methods section, in order to reconstruct the attractor of a chaotic dynamical system with neural networks, it is not enough to minimize the error of short-term forecasts. Instead, one also needs to judge the trained network on its performance for long series of iterated forecasts, and in particular on whether the resulting trajectories resemble those of the original dynamical system. When the training data only covers part of the phase-space, this raises the issue of information availability, as in real-world applications it would not be a valid approach to compare the reconstructed attractor with the full attractor.

All our main experiments were done with feed-forward neural network architectures that forecast the following state of the system. We repeated some of our experiments with networks that forecasted the systems' tendency instead. These were better in producing short-term forecasts in new regions of phase-space, but had even more trouble in producing stable trajectories outside the training space. While feed-forward architectures are widely used, there are many other architectures available, that potentially do not suffer from the issues we found (for example recurrent architectures, echo-state networks and the

related reservoir computers). Chattopadhyay et al. (2019) found that echo-state networks outperform feed-forward architectures in forecasting the Lorenz95 model, and it could be that this also holds for the extrapolation issues addressed in this study. Regarding model architectures, for the forcing experiments it might also be possible that presenting the forcing in another way than done here (e.g. designing into the network that the forcing variable has different characteristics than the state variables) may improve the learning of the influence of the forcing.

To address the second question we raised, we simulated an external forcing on the Lorenz63 and Lorenz95 systems via slowly changing model parameters. We then trained neural networks both with and without the changing model parameters as additional input. Given simulations that span a large enough range of forcing regimes, the networks that use the forcing as input are indeed able to capture at least part of the influence of the forcing, and extrapolate it to some extent to new forcing regimes. The networks again perform better on the Lorenz95 than the Lorenz63 system. This indicates that the idea of emulating climate-change projections with neural networks might not be entirely unrealistic. However, it would be very hard to know beforehand the range of forcing regimes one would need in the training period. Additionally, the networks trained with forcing as an input still perform worse than networks directly trained on the target forcing. Therefore, it may be unwise to apply an architecture that in principle works reasonably on past atmospheric data (like the one proposed by Dueben and Bauer (2018)) to future climates, without very detailed testing. Our results are similar to Rasp et al. (2018), who found that their neural network based a subgrid-model is not able to extrapolate very far into new climate states, even though it is able to interpolate between different extreme climate states. Again, we should highlight that our experiments are not meant to provide a direct match to what may be seen in a climate model. For example, the forcing in the Lorenz63 system is modulated by tuning a parameter that changes the dynamics of the system, while the forcing term in the Lorenz95 system leads to transitions between periodic and turbulent regimes.

More generally, our experiments were performed on highly idealised systems and it is hard to estimate the extent to which they may generalize to more complex systems such as atmospheric general circulation models or even global climate models. Nonetheless, Scher and Messori (2019) have shown that some insights drawn from simple models in the context of machine learning do map to more complex systems. Finally, it is virtually impossible to robustly demonstrate that neural networks cannot fulfill a specific task. In fact, the Universal Approximation Theorem loosely states that a feed-forward neural network can approximate any continuous function with any desired accuracy, as long as it has a large enough number of hidden neurons (Hornik, 1991). However, this does not mean that there is a practically feasible way to *find* the optimal network (network meaning here both architecture and weights) and train it with sufficient data.

We hope that this study can provide a starting point for further discussion on the potentials and limitations of neural networks in the context of chaotic dynamical systems. Future studies could expand to more realistic systems (e.g. general circulation atmospheric models), explore neural network architectures beyond the feed-forward networks used here (e.g. recurrent archi-tectures) and the influence of noisy training data. Additionally, it would be interesting to extend the analysis to the 2-level version of the Lorenz95 model, which would allow to also compare the networks to "truncated" versions of the model. Finally, a more mathematically rigorous approach – as opposed to the empirical approach used here – might shed interesting new light on the topic.

*Code availability.* The code used for this study is available in the accompanying Zenodo repository (doi:10.5281/zenodo.3461683) and on S.S.'s github repository (https://github.com/sipposip/code-for-Generalization-properties-of-neural-networks-trained-on-Lorenz-systems/tree/revision1)

## Appendix A: Tuning of neural network architecture for Lorenz63

The use of neural networks requires a large number of somewhat arbitrary choices to be made before the training of the network
even begins. The first step is to select a specific network architecture, and choose the so-called hyperparameters. As basic architecture here we chose fully connected layers. Next, we performed an exhaustive gridsearch over network configurations and hyperparameters. The learning rate was varied from 0.00003 to 0.003, the number of hidden layers from 1 to 10, and the size of the hidden layers from 4 to 128. The activation function was fixed to the rectified linear unit ("ReLu"). A mini-batch size of 32 was used. The training data was normalized to zero mean and unit variance. The tuning was done with a Lorenz63
run with standard parameters, a timestep of 0.01 and $2e5$ timesteps. While the networks are all trained on short-term error, the final selection of network architecture was done by the ability of the network to reconstruct the attractor (see Section 3.2). The best architecture had 2 hidden layers with a hidden layer size of 128 and a learning rate of 3e-5.

## Appendix B: Tuning of neural network architecture for Lorenz95

For the Lorenz95 model we chose as basic architecture stacked convolution layers, which wrap around the circular domain.
The gridsearch was done over the following parameters: the learning rate was varied from 0.00001 to 0.003; the kernel size of the convolution layers (the "stencil" the convolution operations uses) from 3 to 9; the number of convolution layers from 1 to 9; and the depth of each convolution layer from 32 to 128. Furthermore, both sigmoid and "ReLu" activation functions were tested. A mini-batch size of 32 was used. The training data was normalized to zero mean and unit variance.

The tuning was done with a Lorenz95 run with $F = 8$, a timestep of 0.01 and $1e4$ timesteps. It was performed independently
for forecast lead-times of 0.01, 0.1 and 1. For each lead-time, a different network architecture worked best. When training on lead-times of 0.01, a single convolution layer with kernel size 5 worked best. For a lead-times of 0.1, 2 convolution layers with kernel size 5 worked best, and for a lead-time of 1, 9 convolution layers with kernel size 3 were the optimal choice. When considering how stacked convolution layers work, this result is not surprising. The information available for forecasting the target value for a specific gridpoint is kernel-sized for a single layer, and increases with each additional convolution layer.
From a physical point of view, the information affecting the dynamics of a specific gridpoint comes only from the immediate neighborhood for very short forecasts (given the nature of the Lorenz95 equations). With increasing lead-time the information from an increasingly large part of the domain becomes important. Therefore, it is intuitive that for making a longer forecast in a single step, the network should have more convolution layers.

The network architecture trained on a timestep of 0.1 made the best forecasts over lead-times up to ~4 time units, both in
terms of RMSE and anomaly correlation (when making longer forecasts through iteratively making forecasts with the network, see Fig. B1). We therefore chose this network architecture (conv_depth = 128, kernel_size = 5, learning_rate= 0.003, and 2

convolution layers with ReLu activation) for the analyses presented in the study. This result also suggests that there could be an "optimal" lead-time that neural networks should be trained on for chaotic dynamical systems and is contrary to what Scher and Messori (2019) found on coarse-grained reanalysis data. Indeed, the latter study concluded that the longer the training lead-time, the lower the forecast error. Our architecture only slightly overfits (Fig. B1 c); that is, error on the test data is slightly
5  higher than on the training data. The network was trained until validation loss did not increase for 4 epochs, with a maximum of 30 epochs. The network architecture for the experiments including the forcing $F$ as input was tuned separately. For this, a Lorenz95 run of 1e4 with linearly increasing $F$ from 6 to 7 was used. The last 10 % of the run was used as validation set.

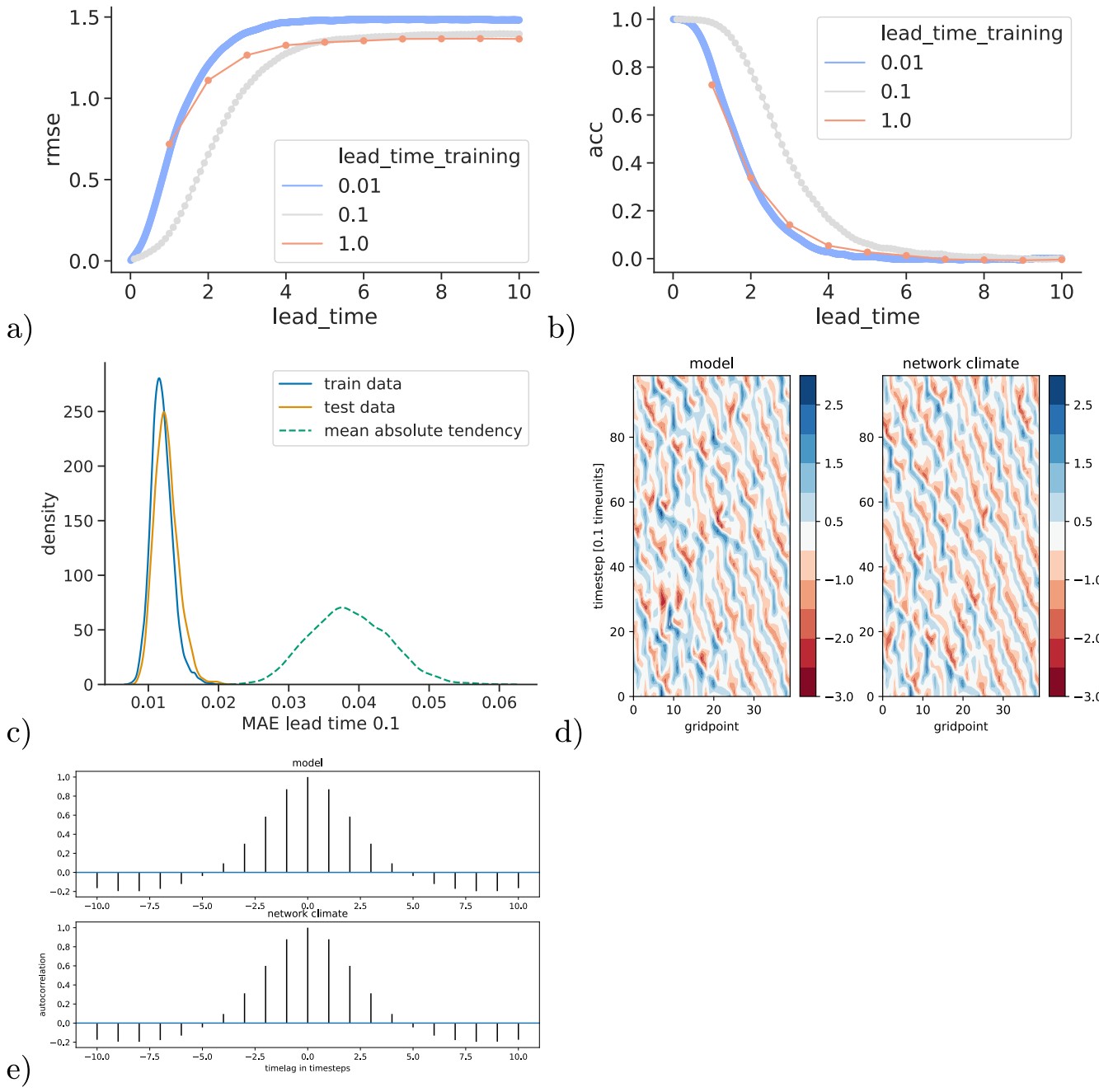

**Figure B1.** Evaluation of network architecture for the Lorenz95 system without $F$ as input. a,b) Forecast error (on test data) for the best network configurations when training on lead-times of 0.01, 0.1 and 1 (different colors). c) Kernel density estimate of mean absolute forecast error on training and test data for 1-step forecasts of the network trained with a lead time of 0.1, and kernel density estimate of the mean absolute 1-step tendencies of the model (dashed line). d) Examples of the Lorenz95 model (left) and the network model obtained through iterated forecasts trained on a lead-time of 0.1 (right), both initialized from the same initial state. e) Autocorrelation for different timelags of the model and the network "climate".

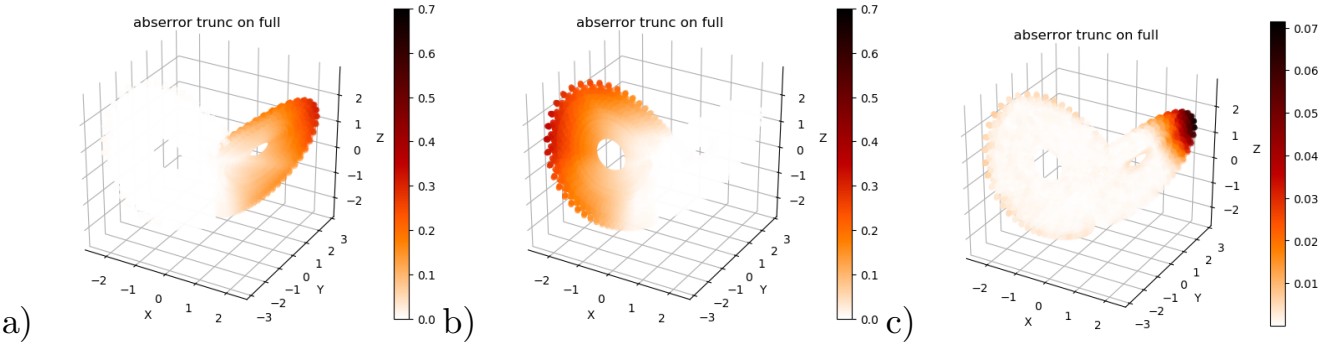

**Figure C1.** Same as Fig. 2 d-f, but for networks forecasting the tendency instead of the following state. Note the different colorscale in c)

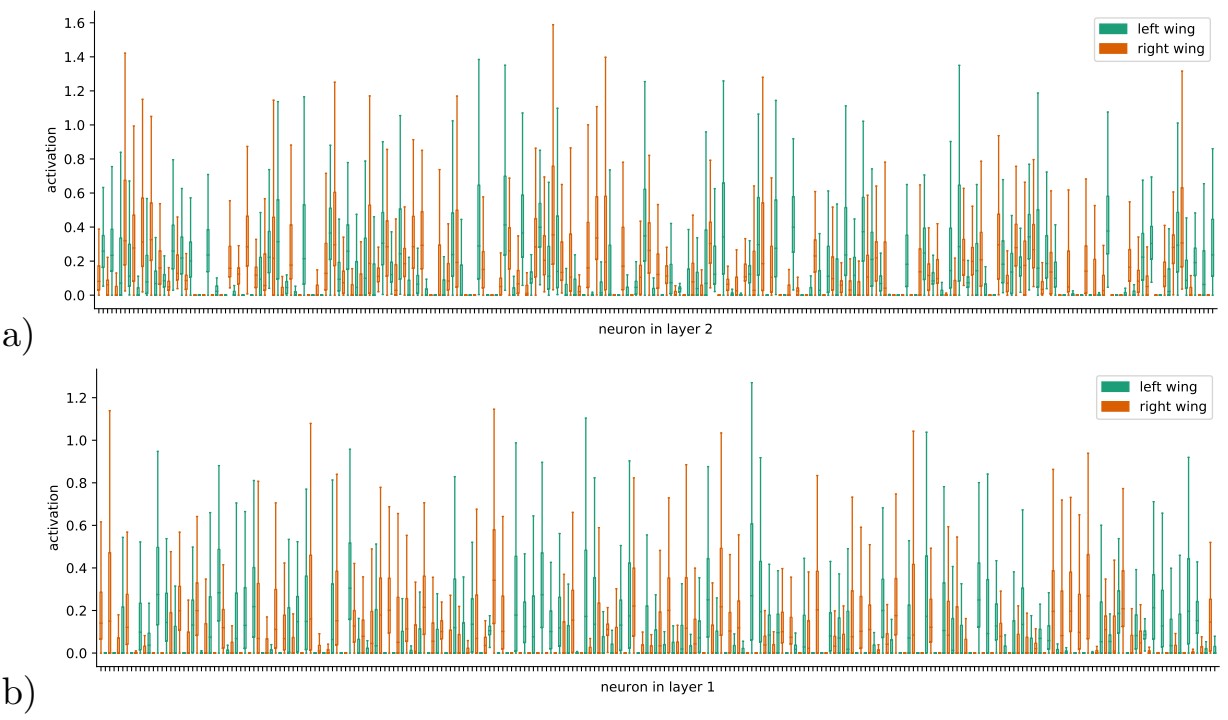

**Figure C2.** Same as Fig. 3b,e, but showing all neurons without specific ordering

## Appendix C: Supplementary figures

*Author contributions.* Both authors developed the ideas underlying this study. S.S. designed the study, implemented the software, performed the analysis and drafted the manuscript. Both authors helped in interpreting the results and improving the manuscript.

*Competing interests.* The authors declare that they have no competing interests.

*Acknowledgements.* S.S. was funded by the Dept. of Meteorology of Stockholm University. G.M. was partly supported by the Swedish Research Council Vetenskapsrådet (grant no.: 2016-03724). The computations were performed on resources provided by the Swedish National Infrastructure for Computing (SNIC) at the High Performance Computing Center North (HPC2N) and National Supercomputer Centre (NSC).

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
