# Peer review of "Generalization properties of feed-forward neural networks trained on Lorenz systems"

_Nonlinear Processes in Geophysics, 2019_

## Referee Comment (RC1) · Anonymous Referee #1 · 24 Jun 2019

**Review of "Generalization properties of neural networks trained on Lorenz systems"**

June 24, 2019

**1   Review**

The paper presents a study of the overfitting properties of feed-forward and 1-d convolutional neural networks and whether they can capture the dynamics along with the influence of an external forcing parameter, in a $40$-dimensional (grid-points of the discretized spatial state) Lorenz-95/96 and a three-dimensional Lorenz-63 system. In Lorenz-63, they show that neural networks struggle to extrapolate the dynamics on attractor regions under-represented in the training data, even in short-term predictions. Long-term predictions starting from these regions do not resemble the ones from the system equations. Moreover, the authors train a small neural network on the whole data and identify neurons of the neural network that are responsible for capturing the dynamics of specific parts in the training data. Thus, they argue that there the neural network learns subnetworks responsible for the dynamics locally and does not learn a global model for the dynamics, which would probably generalize better. In Lorenz-95, they find that including the information of the external forcing variation might degrade

performance and argue that the hypothesis and quest to identify models that work on past climate data, expecting them to work well on future data might be erroneous.

**Quality**

- The quality of the paper is good.

- The authors do not reference related work on generalization properties of neural networks to unseen data, or other machine learning models designed for non-stationary time series.

- They do raise two important open problems in data-driven prediction of dynamical systems. Namely how well neural networks generalize and if they can learn from non-stationarity data.

- The code and data used to generate the results are publicly accessible which enables reproduction of the research.

- The argumentation in Section 2, about whether the network learns only one or many mappings for different regions is inconsistent. The two representations are mathematically equivalent. The impression the reader gets about what the authors are trying to express, is whether different parts of the network are responsible for different (local dynamics) parts of the training data.

- In the Lorenz-63 system, the authors try to answer the aforementioned question (identify specific parts of the neural network that are responsible for capturing the dynamics locally), by analyzing the activation levels of the neurons of the neural network, freezing neurons that are mostly active on specific regions, to check the deterioration of performance on other regions (for a model trained on the whole training data). This argument, that parts of the neural network are responsible for local models of the training data, is very interesting. Especially for large (relative

to the application) overparametrized models, this argument makes sense. However, it has to be tested systematically in larger models (maybe a large model applied to the Lorenz-96), and in a more structured way to be accepted as a general attribute of neural networks, as the small network used in this study might be misleading.

- The authors do not explain the training procedure and how they cope against overfitting in the CNN applied to Lorenz-95. Especially in the low data regime, the absence of measures against overfitting can have a detrimental influence on the performance on the test dataset.

- Since the neural network is forecasting a deterministic system with full state information, the prediction accuracy reported in the Appendix on page 17, seems quite low. In the provided plots, the networks seems to be forecasting inaccurately, as the difference in the plots even at early timesteps is obvious.

- The generalization of the study is problematic. The study is limited to feedforward neural networks, with a one to one training scheme. By changing the loss to include some stability metric, or long-term performance, trying out different architectures, regularization techniques, etc. the generalization properties might be improved. The first problem of extrapolation is a general pitfall of data-driven approaches, however, the second problem of non-stationarity might be alleviated with more sophisticated architectures. As reported in Section 3.4, many trained CNN networks are not stable, because they were trained for single step forecasts. This is expected, as the neural networks are not trained for long-term forecasting. RNNs can be used in these low-dimensional systems, backpropagating the gradient many timesteps in the past to ensure stability. The authors use a posteriori analysis of the networks to identify the stable ones. Moreover, the models are applied to non-stationary timeseries with external forcing, which is a really challenging application. The selected models and the training procedure used is not

adequate to extract general conclusions. For example, complex RNN architectures that try to capture multiple time scales, or Reservoir Computing approaches might work better. The conclusions of the paper should be specific only to feedforward neural networks. The arsenal of machine learning tools to counter these open problems is much wider.

- The second question the paper poses, is a very interesting one. Real time-series data are most of the time non-stationary. Even though many neural architectures have been successfully used in seasonal or non-stationary data, it is not clear if the networks can actually learn varying dynamics, or how efficient they are in that. One solution could be to train networks on the fly as new data come in. There is available literature on applying machine learning approaches for non-stationary data. Even though the model used in this study appears to be incapable of generalizing, this might not hold for other models. For example, the forcing was provided as an additional input to the network. However, we do know that this external forcing is not the same type of input as the rest. This information could be provided in a different way to the network.

- The statement "... the trajectories of the network forecast simply point back towards the region included in the training." regarding the behavior of the neural network in regions of the phase space not included in the training data, seems rather arbitrary. Since the neural network is not trained in these regions the behavior can be anything.

**Clarity**

- Good. The results of the paper and the conclusions are clearly explained.

**Originality**

- The first problem of generalizing to unseen data is a well-known one. As a data-driven approach, neural networks have a hard time to extrapolate to unseen regions in the dataspace. This is addressed in many previous studies, not only related with dynamical systems. Regularization, coupling neural networks with equations, adding constraints etc. are known measures to cope with this deficiency. Most data-driven methods suffer from this problem. It is not surprising to see that a small neural network trained on the left wing of the Lorenz-63 attractor cannot generalize to the vastly different dynamics (in terms of data, not equations) of the right wing. The situation is expected to worsen as the models grows bigger (overfitting easier).

- Implicitly, the authors state a very interesting question, whether the neural networks learn sub-networks that are responsible for modeling the dynamics locally in parts of the training data. In the Lorenz-63 system, they manage to demonstrate this in terms of identifying the neuron that seems to be responsible for a specific part of the training data (right or left wing). Whether this argument holds for large models or other more complicated dynamics and is not specific to this study, remains open. However, it is an original and interesting finding that needs to be tested for more general settings (large networks, more applications).

- The second issue raised, is whether NN can learn dynamics that evolve based on external forcing. This is connected with the known open problem of neural networks learning from non-stationary data/dynamics. The architectures proposed in the study are not compared with other state-of-the art approaches, like reservoir computers, RNNs, ARIMA models, etc. and long-term results are not presented (from iterative forecasting) so it is not straightforward to judge their efficiency.

**Confidence**

- The reviewer is confident but not absolutely certain.

**Recommendation**

- Accept subject to major revisions.

- Accept after the revision of the issues raised above, or at least referencing them in the text. Especially for the argument about the sub-networks having learned local dynamics, a bigger model needs to be tested. I doubt there is any model applied in practice with only 8 neurons. ML models applied in practice have thousands to millions of parameters. In order to support this claim, it has to be tested on large models in a systematic way, which is however, challenging to achieve.

**2  Typos**

1. Page 1, line 18, typo "... the widely studied ..."

2. Page 2, line 5, typo "... the widely In in this paper ..."

3. Page 4, line 24, typo "The neural networks are trained..."

4. Page 6, line 19, In order to avoid misconception, the following reformulation would help the reader: "Figure 2 shows the training data and the forecast error for a network ..."

5. Page 15 line 3, typo "... of how to test test the reconstruction at training time ..."

---

## Short Comment (SC1) · 26 Jun 2019

Sebastian Scher

sebastian.scher@misu.su.se

Dear Reviewer,

Thank you for your thorough and very constructive review. We will write a detailed reply after receiving the comments from the other reviewers. However, to clarify a few points and to aid the other reviewers, we briefly reply to your main points and outline how we plan to address them.

You are absolutely right in that all conclusions from our paper do only apply for feed-forward networks. We realize that we should have pointed this out already in the abstract and potentially in the title. In the revised version, we will mention this very clearly, and also discuss it more verbose in the main text/conclusion. Additionally, we plan to

extend the analysis of neuron-activations to larger neural network models.

*"The argumentation in Section 2, about whether the network learns only one or many mappings for different regions is inconsistent. The two representations are mathematically equivalent. The impression the reader gets about what the authors are trying to express, is whether different parts of the network are re-sponsible for different (local dynamics) parts of the training data"*

This is indeed what we wanted to express. We agree that the mathematical notation might be misleading, and we will either remove it or explain it better in the revised version.

*"The authors do not explain the training procedure and how they cope against overfitting in the CNN applied to Lorenz-95. Especially in the low data regime, the absence of measures against overfitting can have a detrimental influence on the performance on the test dataset."*

Thanks for pointing out that we forgot to include the exact training procedure of the CNN for the Lorenz95 in the text. We used the last 10 percent of each training set as validation data, and controlled overfitting via monitoring validation loss (the training is stopped when the validation loss has not improved for more than 4 epochs, or when 30 epochs were reached.) We will explain this more clearly in the revised version.

*"Since the neural network is forecasting a deterministic system with full state in- formation, the prediction accuracy reported in the Appendix on page 17, seems quite low. In the provided plots, the networks seems to be forecasting inaccu- rately, as the difference in the plots even at early timesteps is obvious."*

It is true that the Lorenz95 system is a deterministic system, but it is a chaotic deterministic system. In fact it was explicitly designed in order to have chaotic behaviour for the study of predictability-limits (see Lorenz 1996). For the lorenz95, this predictability limit is at a forecast-time of roughly 2-3 time-units. This is usually expressed in terms

of initial condition uncertainty (if there is a very small error in the initial conditions, after reaching the predictability limit, the forecast will be only as good as a random forecast). However, it also translates to model uncertainty: if a surrogate-model of the system is not absolutely perfect, it will not be able to make good forecasts behind the predictability limit. Therefore, it is expected that any type of surrogate model (like a neural network) won't be able to do well after this predictability limit (which is exactly what we see). This is also seen in other studies were the error of neural-network forecasts on the lorenz95 system increases rapidly with forecast-time (e.g. Vlachas et al 2018). We do however realize that this is quite un-intuitive for readers not familiar with atmospheric predictability studies, and we will add discussion on this in the revised manuscript. Additionally, we now realize that panel c) of figure A1 might be misleading, and might have caused your comment that *"In the provided plots, the networks seems to be forecasting inaccurately, as the difference in the plots even at early timesteps is obvious."* The plot shows the evolution of the lorenz95 model and the network climate, however they were not initialized with the same state. The plot was intended not to show forecast-performance, but to demonstrate that long runs of the CNN do look realistic. However, this is not obvious from the caption, and we apologize for the confusion. In the revised version we will show plots of runs that are actually initialized from exactly the same state, then the plots can be used to analyze forecast performance as well. We will also make this clearer in the caption.

Lorenz, E. N.: Predictability: A problem partly solved, in: Proc. Seminar on predictability, vol. 1, 1996.

P. R., Byeon, W., Wan, Z. Y., Sapsis, T. P., and Koumoutsakos, P.: Data-driven forecasting of high-dimensional chaotic sys-tems with long short-term memory networks, Proc. R. Soc. A, 474, 20170 844, https://doi.org/10.1098/rspa.2017.0844, http://rspa.royalsocietypublishing.org/content/474/2213/20170844, 2018

*"The statement "... the trajectories of the network forecast simply point back to- wards the region included in the training." regarding the behavior of the neural network in*

*regions of the phase space not included in the training data, seems rather arbitrary. Since the neural network is not trained in these regions the be- havior can be anything."*

What we meant to say is that we actually observe that in our trained networks, the network forecasts initialized outside the training phase space do point back to the training phase space. You are of course right that as an a-priori assumption this would be rather arbitrary. We will make it more clear in the revised version that this is an empirical observation.

Sebastian Scher

---

## Referee Comment (RC2) · Anonymous Referee #2 · 25 Jul 2019

**Review of "Generalization properties of neural networks trained on Lorenz systems" by Scher and Messori**

Note I wrote these comments before viewing those from the first set of review comments.

This paper addresses the questions of whether neural networks can:
1. learn to simulate dynamical systems if the training data does not include the full range of possible system states and
2. learn to project the effect of a changing forcing on a dynamical system into the future, as the forcing increases beyond the range seen in training.

This is done with experiments using two fairly simple dynamical systems. The machine learning algorithms do not perform very well at either task overall and the authors argue that this illustrates challenges for machine-learning applications. It is suggested that the results have relevance for applying machine learning to simulate the climate system.

My overall opinion is that whilst the questions the paper addresses are interesting and important, and the results are well-presented on the whole, the experiments performed are not very close to how neural networks would be applied in reality, so it's not clear if they have real-world applications (even notwithstanding the simplicity of the systems being studied). In particular, the performance of the neural networks on reproducing the training data often looks so poor that they would not be used in an application, or the training performance is not presented in enough detail, so it's not clear that the results would apply to real-world applications that would require good validation performance. Also, the changes in forcings applied in the second part seem a lot larger than for applications that neural networks might be considered for. In addition, the authors' experiments on the Lorenz '63 system have used a particular neural network design, where the full state is predicted at every time step (it's not clear if this is the case for the Lorenz '95 system as well) – this may be expected to work worse than other designs where only the change in the state is predicted at each step, or where bias-correction of an approximate dynamical model is performed, and the results here are not clearly generalisable to those set ups.

I have given more detailed comments below. I think the work could eventually be publishable, if the comments are adequately addressed. Since my comments are quite substantial, it could be acceptable to just include the L63 experiments on training on part of the attractor and the L95 experiments on response to forcing – the L63 experiments on responding to a parameter change seem less applicable to real-world cases like predicting climate change. I do encourage the authors to continue with this line of investigation, which I think is potentially very valuable.

**Most significant comments:**
1. Training performance of neural networks:
   a. For the Lorenz '63 experiments, the ability of the trained neural network to reproduce the attractor appears quite poor (fig.1), and much worse than in the results presented by Zhang (2017), whose work the authors say they are following. A model with such performance would not be used in any real-world application I can think of, and the later results may be much worse than for a well-trained system. It should probably be checked that all of the important steps in the prior work were followed, and if that does not resolve the problem, different architectures tried (e.g. using more neurons) until a good simulation of the attractor is produced.

b. More diagnostics for the performance of the Lorenz '95 networks on the training data and the equivalent portion of the test data should be presented to indicate the system's performance and the degree of overfitting e.g. MAE of single-timestep predictions relative to the variance of the system's tendencies.

c. In the experiments testing how well the networks capture the response to a changing forcing, it needs to be shown how well the networks reproduce the trend in the training data. For the results to be applicable to predicting the path of global warming, for example, there should be a discernible trend in the training data and the neural networks should reproduce it with an accuracy similar to what climate models achieve, else they would be deemed to be unsuitable for use in prediction.

2. Forcing experiments:

a.
- The changes in the forcing terms are rather large compared to the effects expected from anthropogenic climate forcing, for example, and I'm not aware of another real-world case where neural networks would be considered for modelling the effects of such large changes in forcing, so it's not clear to me that these results have real-world applicability. For context, anthropogenic radiative forcing of the climate system is projected to be up to a few percent of solar radiative forcing, and in the scenarios with the largest climate changes, total global warming is around 5x what has been seen in the $20^{th}$ century, and comparable in size (though not rate) to changes in between ice ages (so we arguably have some data for testing whether models can simulate such large changes well). In the Lorenz '63 experiments here, the change in the sigma parameter (meant to be analogous to radiative forcing of climate?) changes by a factor of 2. In the Lorenz '95 experiments, the forcing change is enough to change the system from being periodic to highly turbulent, which is a much larger qualitative change than expected from climate change. I wouldn't have expected neural networks to perform well at the tasks set, namely simulating systems that are very different from what they've been trained on, so these results don't seem to provide much new information.
- It seems reasonable to think that neural networks could perform better at simulating the effects of smaller forcing changes, that are more comparable to those in real situations. It would be interesting to test whether the neural networks can reproduce the effects of forcing at the level seen at the end of their training period (relevant for attributing observed weather events to climate change, for example e.g. National Academy of Sciences, 2016, "Attribution of Extreme Weather Events in the Context of Climate Change") and if so, how far beyond the range of forcing they were trained on can they make good predictions for? (c.f. the Paris climate agreement global warming targets of 1.5C and 2C, which are ~1.5x and ~2x the observed warming – it would be interesting to know if neural networks could provide results that are at all useful for predicting the effects of forcing changes of that magnitude.)
- As a further comment, it doesn't seem likely that neural networks could learn the effects of forcings outside the range of the training data without having additional information about the effects of larger forcings e.g. the radiative effects of CO2 in the climate change context. So it seems *a priori* likely that for the given setup the performance will deteriorate as the forcing becomes larger. Perhaps the experiments here could demonstrate this, but I don't think it would be that surprising.

b. The finding that including information about the forcing as an input often worsens performance seems surprising. One reason could be that the network architecture was tuned to optimise performance without the forcing input, and a larger architecture may be needed to perform well with this information. To be a fair test, the network architecture search should be repeated for the networks using forcing as an input – this may be especially relevant for the L63 case, where the network used is quite small. The

statement in the discussion that "it may be better not to include the forcing variable as network input" does not seem well-justified due to this, and also because I do not see how in principle a neural network could predict the effect of a change in forcing if it is not given information about the forcing.

3. It should be made clear in the abstract and conclusions that the results apply for a particular choice of neural network design, namely feedforward networks predicting the system state at time t+1 given the state at time t (it's not clear if this is also the case for the L95 experiments, and this should be clearly stated). Also, the L63 experiments testing whether neural networks could represent the system in one wing of the attractor having been trained on the other wing only used sigmoid activation functions.

   • Predicting the whole state at every time step may be expected to work worse than other designs where only the change in the state is predicted at each step (e.g. Dueben and Bauer, 2018), or where bias-correction of an approximate dynamical model is performed (Watson, 2019, https://doi.org/10.1029/2018MS001597). This is because in these cases, a lot of the variance in the quantity being predicted is removed, so minimising the RMSE in training may work better to give a system that is capturing the important aspects of the variability. These different methods should also be discussed, and the abstract and conclusions should say that the results may not apply to methods like these.

   • The choice of sigmoid activation functions for the L63 network may be relevant for the result that the network will not make predictions outside of the range of its training data because sigmoids saturate, and may have trained to saturate at prediction values that are not far outside the boundaries of the training data region, making it difficult for the neural network to predict values outside this region. It would be good to check what happens when using an activation function that does not saturate e.g. ReLu. (Though I still wouldn't expect it to work well when so much data is left out from training – but I'd still find the result interesting, particularly if done with a system that predicted the tendency rather than the whole state).

**Other comments**
1. p.1 L19 It would seem relevant to include citations to other recent studies using neural networks to simulate the Lorenz 95/6 system (Chattopadhyay et al., 2019, https://doi.org/10.31223/osf.io/fbxns; Watson, 2019, https://doi.org/10.1029/2018MS001597).
2. p.2 L10 - Perhaps also mention Lorenz '95 is sometimes called Lorenz '96
3. p.2 L10-12 – Some context here may be useful. For paleoclimate variability and the oceans over multiple decades, yes, but it's less likely to be the case for the atmosphere that unforced variability would be far outside what we've observed.
4. p.3 L9-10 Training where no large regions of phase space is left out seems to be the most realistic case for atmospheric modelling, which is what is referred to. The experiments may be relevant for e.g. ocean modelling, where time scales are much longer. (I do think they are inherently interesting, as well.)
5. p.3 L25 I'm not sure if all readers would be familiar with the Lorenz butterfly - perhaps refer to a figure.
6. p.3 L26 A better description of the solver is needed e.g. what software package is this from? Reference?
7. p.4 L1 Lorenz95 seems to be more often used to describe the 2-level model Lorenz introduced in the same paper. Perhaps use a different name to make it clear you are considering the 1-level version. (As an aside, the 2-level model could be used to test how well neural networks perform compared to a "truncated" model of the system i.e. the 1-level model – this would address whether neural networks can improve upon other reasonable dynamical models, which is a key question.)

8. p.4 L4 What is the behaviour of the system like with N=40? A plot of a time series or similar may be helpful.
9. p.4 L8-12 More detail seems necessary to make the results reproducible - e.g. Were input variables normalised? Regularisation? Minibatch size? Learning rate? Stopping rule?
10. p.4 L21-22 Why not give F as one value in the L95 system? (Analogous to using $CO_2$ concentration as an input to a climate model.)
11. p.5 L8 – how big are the grid boxes being used?
12. p.5 L8-9 – what does "normalized data" mean here?
13. p.5 L22 For testing 1), the trajectory will tend to the fixed point and only reach it after infinite time, so it's not clear that it could be identified by looking for two points to be exactly equal - perhaps set a threshold on how small a tendency is acceptable and check if the tendency magnitude falls below this and does not rise above it again?
14. p.5 L23-24 What is the motivation behind 2? To identify periodic behaviour? But couldn't points on a periodic orbit fall at slightly different points on that orbit and so avoid detection?
15. p.6 L1-2 The information on training here could perhaps be merged with the earlier section about training.
16. p.6 L5 what does "consecutive forecasts" mean as opposed to just "forecasts"?
17. p.6 L5-6 See earlier comment about the simulated attractor not looking reasonable.
18. p.6 L9 How big are these errors compared to an average tendency magnitude? Also, could errors on the wings be larger because tendencies are typically larger? Perhaps showing a relative error would be more useful.
19. p.6 L14-15 I didn't get the meaning of "the points...included point".
20. p.8 L11-12 The errors look quite a bit smaller to me, particularly for the case of fig.3f..
21. p.8 L12-14 I don't feel convinced by this analysis. Even if the neural network had learnt to fit the whole attractor well, there may be some neurons whose activations only vary in part of the phase space due to the values of the inputs, and fixing their values would of course be expected to degrade forecasts in areas of the phase space where they do matter.
22. p.11 L5-6 This experiment is not really like rising anthropogenic climate forcing because the sign of the "forcing" depends on whether y is larger than x. An additive forcing term like that used for the L95 experiments and by Palmer (1999, "A nonlinear dynamical perspective on climate prediction", J. Clim.) would be a better analogy.
23. p.11 L6-7 Changing the number of time steps looks to change the rate of change in the forcing as well. This is something that should be made clear. The results will conflate the effects of changing the amount of training data and the rate of change of the forcing. Perhaps a test could be done for the case with a lower rate of change of forcing with the lower number of time steps used in training to see the effect of changing each aspect of the set up.
24. p.11 L17 It's not clear to me if different training and testing runs are produced for each experimental repeat.
25. p.13 L2 Given that the performance of the Lorenz systems with fixed F also vary between the situations with different training data lengths, could the different rates of forcing change in each case be affecting the system's behaviour in some important way?
26. p.15 L25 - "layers" should be "neurons"?
27. fig.2 - It should be pointed out clearly that the colour scales are different.
28. fig.3 - It's a bit unclear to compare the distributions across panels b and e to see which neurons change behaviour more - it might be better to put the values side by side on the same axes.
29. fig.3 – panel references are wrong in the caption. It also needs to be clearer about what the distributions in b and e are.
30. fig.5 - It needs to be made clear that single-timestep prediction errors are being shown. It would be more interesting to see metrics of longer-range forecast skill and how well the

attractor is simulated, since in the end single-timestep predictions are not the target and may not indicate that well the performance on longer time scales.

31. fig.5a - I suggest using a different colour for the vertical lines.
32. fig.5 - It would be better to write "Ntrain" in exponential notation.
33. fig.5 - The orders of the Lorenz systems could be reversed so they go start to end, the same as for the neural networks.
34. fig.5 - There are no bars on the simulations with the Lorenz system, so have you used the same Truth runs for each experiment, so you could only run the Lorenz systems once, or are the bars just not visible? It would be good to make small bars visible by giving their ends a width so they stick out.
35. References – some references have repeated DOIs.

---

## Referee Comment (RC3) · Anonymous Referee #3 · 31 Jul 2019

Summary:

This paper investigates the very topical question to what extent neural network models can be generalized in the context of dynamical systems. The authors use two well-known models which display chaotic behaviour: the Lorenz 63 and Lorenz 95 models. Two aspects are addressed. The first aspect is the representativity of a neural network which is trained on a severely limited set of training data from the Lorenz 63 model (in this case, the removal of one entire "wing" of the Lorenz butterfly, or just the tip of one wing). The second aspect relates to the representativity of a neural network model under a changing parameter or forcing, when this parameter is provided as input. These parameters are the sigma parameter in the Lorenz 63 model, and the forcing parameter F in the Lorenz 95 model.

[Figure]

The neural network is evaluated not only by its short-term predictive skill but also by its ability to reproduce the long-term characteristics of the original model. Three different metrics are proposed, the so-called density-selection and density-full approaches, and a third approach which rejects neural network models with fixed points or periodic solutions.

The authors show that indeed, selectively removing half the training data will lead to a NN model which performs very poorly outside the wing on which it was trained. They interpret this as follows: the NN model approximates a local, and not a global function. They try to pinpoint the source of this locality by looking at the spread of the neurons' activations.

Likewise, training the model for a certain range of sigma values does not yield a good model outside this range, even if sigma is provided as an input parameter. There is no clear benefit from including the forcing parameter F as an input parameter in the analogous experiment in the Lorenz 95 model.

General comments:

Given the recent enormous successes of machine learning, it's only natural that this approach is being adopted enthusiastically in many different branches of science. This makes it all the more important to highlight the limitations of machine learning methods in disciplines such as climate science. I applaud the authors' effort to provide a tangible example of where the NN approach breaks down and to investigate what the causes are.

The paper is easy to read and the results are presented in a clear and well-structured way. The reproducibility and transparency of the results are supported by the availability of the code on Zenodo. Overall the manuscript is of a high quality. As for the novelty of the results and the context, however, I have some remarks that may require a substantial revision of the manuscript.

A first remark is that the first experiment seems quite artificial. The Lorenz 63 model is an extremely idealized model with a very peculiar bifurcation structure and distinct symmetries, making it less than ideal to represent a realistic general circulation model. A more realistic model which also displays regimes such as that of Charney and Straus (1980), would probably have been more suitable for this particular experiment. Of course, the authors mention these caveats, but in the end there are so many caveats that it seems that no conclusion can be drawn at all related to realistic climate models. Moreover, it is not because some insights related to machine learning map to more complex models, that this is a universal property. I'm not sure how the authors can address this issue without performing the analysis for a more realistic model. Nevertheless, I appreciate the value of this experiment, albeit in a more theoretical context of dynamical systems theory.

Secondly, the result of the forcing experiment of the Lorenz 95 model seems hardly surprising, as the forcing parameter is varied from the periodic regime to the turbulent regime. One cannot expect a neural network to predict qualitatively completely different behaviour.

Finally, it seems to me that similar studies must have been performed in the literature on neural networks, though not necessarily in the context of geophysics. I would encourage the authors to explore the literature on this. The recent groundbreaking success of deep learning was only possible thanks to the move from few to many hidden layers, and it appears that large deep learning networks have better generalization properties than smaller ones. It would therefore also be interesting to repeat the exercise for a deep neural network. See for example the work by Novak et al. (2018) or Wu et al. (2017) who investigate the source of these generalization properties, and references therein.

Specific comments and typographical errors:

p.1, L 9-10: In the abstract, the authors conclude that "These results outline challenges

for a variety of machine-learning applications. [...]". The word "outline" (in the sense of summarize) goes a bit too far since the results shown are for two highly specific models and a very artificial set-up (an entire wing missing, training in periodic regime). I would just say that the results provide some examples.

Figure 3: Labels in the caption don't match with the relevant subfigures.

p. 11, L 16: lorenz -> Lorenz (2x)

p. 15, L 1: However, also the alternative methods suffer -> However, the alternative methods also suffer

p. 15, L 3: test test -> test

Check spelling consistency: throughout the manuscript, generalize / generalise are both used

References:

Charney J G, Straus DM (1980) Form-Drag Instability, Multiple Equilibria and Propagating Planetary Waves in Baroclinic, Orographically Forced, Planetary Wave Systems. J Atmos Sci 37: 1157-1176.

Roman Novak, Yasaman Bahri, Daniel A. Abolafia, Jeffrey Pennington, Jascha Sohl-Dickstein (2018) Sensitivity and Generalization in Neural Networks: an Empirical Study. arXiv:1802.08760.

Lei Wu, Zhanxing Zhu, Weinan E (2017) Towards Understanding Generalization of Deep Learning: Perspective of Loss Landscapes. arXiv:1706.10239.

---

## Author Comment (AC1) · 12 Aug 2019

We thank the reviewers for their constructive comments. As we understand, the main criticisms shared by all reviewers were on the interpretation of our results, and whether the experiments are realistic enough to have real-world relevance. We outline here how we plan to address the main points raised by the reviewers. We will include a detailed point-by-point response when we submit our revised manuscript.

We are planing to:

- Do an extensive network-tuning search for the Lorenz63 model, to objectively find a good architecture. Specifically, we will explore larger (and deeper) neural network models in order to test whether our results also hold for larger models.

- Add more validation metrics for the network used on the Lorenz95 model

- Make it clearer in the manuscript that all results do only apply to feed-forward neural networks

- We argued that the networks do in fact learn sub-networks for different parts of the attractor. We will extend this analysis to more complex networks.

- Redesign our forcing experiments on the Lorenz95 system. Specifically, we will also test whether the network is able to extrapolate to situations just out of the training regime, in contrast to the quite extreme experiment included in the initial submission.

- Repeat the tuning of the network for the Lorenz95 system also with forcing as input, in order to allow a fair comparison.

- Consider expanding at least part of the analysis to networks that do not predict the whole state, but that predict the tendency instead.

- Better contextualize our work relative to the broader literature on generalization of neural networks

- Extend the analysis on generalization to different phase-space regions to the Lorenz95 model as well which, although highly idealized, is more pertinent to real climate models than the Lorenz63 system.

---

## Author Response (AR1)

We thank the reviewers for their constructive comments. We have now prepared a new manuscript based on their comments. The main changes we made are:

- the network architecture for the Lorenz63 model is now based on objective tuning, and is more similar to real-world architectures than the one in the original manuscript
- the analysis of extrapolation capabilities of the neural networks is now also done on the Lorenz95 model, in addition to the Lorenz63 model
- the forcing experiments have been re-designed to be more close to real applications
- the analysis of the importance of different regions of the neural network has been extended to several different architectures

As you will see, thanks to these new experiments, the main result of the paper slightly changed. While the result that the neural networks do not extrapolate to new phase-space regions in the Lorenz63 model still holds, for the Lorenz95 this does not seem to be the case. Also, the new forcing experiments show that the networks are to some extent learning external forcing, however only when given large ranges of forcing in the training.

We provided point-by-point response to all reviewers' comments below (in red).

**Reviewer #1**

The authors do not reference related work on generalization properties of neural networks to unseen data, or other machine learning models designed for non-stationary time series.

We have added a new section in the introduction ("1.2 Related work on generalization properties of neural networks") that gives a better overview of the general literature on generalization properties of neural networks.

The argumentation in Section 2, about whether the network learns only one or many mappings for different regions is inconsistent. The two representations are mathematically equivalent. The impression the reader gets about what the authors are trying to express, is whether different parts of the network are responsible for different (local dynamics) parts of the training data.

Yes, this is indeed what we had meant. Even though the two expressions are mathematically the same, we think that they help explaining this idea. Therefore, we added "Even though mathematically equivalent, the latter would imply that different parts of the network are responsible for different regions of the phase-space." to make this clearer. (p3 L15)

In the Lorenz-63 system, the authors try to answer the aforementioned question (identify specific parts of the neural network that are responsible for capturing the dynamics locally), by analyzing the activation levels of the neurons of the neural network, freezing neurons that are mostly active on specific regions, to check the deterioration of performance on other regions (for a model trained on the whole training data). This argument, that parts of the neural network are responsible for local models of the training data, is very interesting. Especially for large (relative to the application) overparametrized models, this argument makes sense. How-ever, it has to be tested systematically in larger models (maybe a large model applied to the Lorenz-96), and in a more structured way to be accepted as a general attribute of neural networks, as the small network used in this study might be misleading.

In the new manuscript, we use a larger network throughout for the Lorenz63 (2 hidden layers with 128 neurons each). Additionally, we extended the analysis regarding neuron activations to a wider range of architectures. (fig. 4 in the new manuscript).

The authors do not explain the training procedure and how they cope against overfitting in the CNN applied to Lorenz-95. Especially in the low data regime, the absence of measures against overfitting can have a detrimental influence on the performance on the test dataset. Since the neural network is forecasting a deterministic system with full state information, the prediction accuracy reported in the Appendix on page 17, seems quite low. In the provided plots, the networks seem to be forecasting inaccurately, as the difference in the plots even at early timesteps is obvious.

We now added more detail on the training procedures both in the method section and in the Appendix. Additionally, we changed the plot you referred to. The wave-plot in the original manuscript showed the evolution of the real system, and the evolution of a long run of the network. However, they were not initialized from the same state, therefore giving the impression that the forecasts are very bad. The plot in the new manuscript (fig. B1 c) now shows the same, but with the network run initialized from the same initial state as the real system. From this, together with the other panels of fig. B1, it can now be seen that the network forecasts are skillful.

The generalization of the study is problematic. The study is limited to feedforward neural networks, with a one to one training scheme. By changing the loss to include some stability metric, or long-term performance, trying out different architectures, regularization techniques, etc. the generalization properties might be improved. The first problem of extrapolation is a general pitfall of data-driven approaches, however, the second problem of non-stationarity might be alleviated with more sophisticated architectures. As reported in Section 3.4, many trainedCNN networks are not stable, because they were trained for single step forecasts. This is expected, as the neural networks are not trained for long-term forecasting. RNNs can be used in these low-dimensional systems, backpropagating the gradient many timesteps in the past to ensure stability. The authors use a posteriori analysis of the networks to identify the stable ones. Moreover, the models are applied to non-stationary timeseries with external forcing, which is a really challenging application. The selected models and the training procedure used is not adequate to extract general conclusions. For example, complex RNN architectures that try to capture multiple time scales, or Reservoir Computing approaches might work better. The conclusions of the paper should be specific only to feed-forward neural networks. The arsenal of machine learning tools to counter these open problems is much wider.

We completely agree that our results are specific to feed-forward architectures only. We realize that we should have made this clearer in the original manuscript. In the new manuscript, we now clearly mention this in the abstract and the conclusion section. Also, we changed "neural networks" in the title to "feed-forward neural networks". Additionally, we discuss in the conclusion section that other methods might alleviate the problems we found.

The second question the paper poses, is a very interesting one. Real time-series data are most of the time non-stationary. Even though many neural architectures have been successfully used in seasonal or non-stationary data, it is not clear if the networks can actually learn varying dynamics, or how efficient they are in that. One solution could be to train networks on the fly as new data come in. There is available literature on applying machine learning approaches for non-stationary data. Even though the model used in this study appears to be incapable of generalizing, this might not hold for other models. For example, the forcing was provided as an additional input to the network. However, we do know that this external forcing is not the same type of input as the rest. This information could be provided in a different way to the network.

You are right that the forcing is not of the same type as the rest of the input. We added the following sentence in the conclusion: "Regarding model architectures, for the forcing experiments it might also be possible that presenting the forcing in another way than done here (e.g. designing into the network that the forcing variable has different characteristics than the state variables) may improve the learning of the influence of the forcing." (p20 L3) We have further significantly updated the section discussing varying forcing, for both the Lorenz63 and Lorenz95 attractors.

The statement "... the trajectories of the network forecast simply point back towards the region included in the training." Regarding the behavior of the neural network in regions of the phase space not included in the training data, seems rather arbitrary. Since the neural network is not trained in these regions the behavior can be anything.

What we meant to say is that we actually *observe* that in our trained networks: the network forecasts initialized outside the training phase space do point back to the training phase space. You are of course right that as an a-priori assumption this would be rather arbitrary. We now removed the word "simply" to make clear that this is not simply a trivial a-priori assumption, but an observation.

The first problem of generalizing to unseen data is a well-known one. As a data-driven approach, neural networks have a hard time to extrapolate to unseen regions in the dataspace. This is addressed in many

previous studies, not only related with dynamical systems. Regularization, coupling neural networks with equations, adding constraints etc. are known measures to cope with this deficiency. Most data-driven methods suffer from this problem. It is not surprising to see that a small neural network trained on the left wing of the Lorenz-63 attractor cannot generalize to the vastly different dynamics (in terms of data, not equations) of the right wing. The situation is expected to worsen as the models grows bigger (overfitting easier).

We now use a bigger network as our main architecture, and the situation is very similar to the architecture in our original manuscript. We agree that the example of training on one wing and testing on the other wing is quite "extreme" – which we indeed mention in the text – but we think it provides a good example that the network does in fact not learn the underlying equations.

Implicitly, the authors state a very interesting question, whether the neural networks learn sub-networks that are responsible for modeling the dynamics locally in parts of the training data. In the Lorenz-63 system, they manage to demonstrate this in terms of identifying the neuron that seems to be responsible for a specific part of the training data (right or left wing). Whether this argument holds for large models or other more complicated dynamics and is not specific to this study, remains open. However, it is an original and interesting finding that needs to be tested for more general settings (large networks, more applications).

We agree that in order to generalize our result, one needs to test different networks, and also different applications. To address the first point, we have now extended our analysis to a wider range of network architectures, training from shallow ones with few neurons, to deeper ones with wide layers. The finding of different parts of the network controlling different regions of phase space also – at least to some extend – holds for these architectures (see section 4.1.2). While it would indeed be very interesting to also look at other applications, we think this is beyond the scope of this study. However, in the conclusion section we now reference to a study in the field of image generation with Generative Adverserial Networks that had similar results (https://arxiv.org/pdf/1811.10597.pdf)

The second issue raised, is whether NN can learn dynamics that evolve based on external forcing. This is connected with the known open problem of neural networks learning from non-stationary data/dynamics. The architectures proposed in the study are not compared with other state-of-the art approaches, liker reservoir computers, RNNs, ARIMA models, etc. and long-term results are not presented (from iterative forecasting) so it is not straightforward to judge their efficiency.

Accept after the revision of the issues raised above, or at least referencing them in the text. Especially for the argument about the sub-networks having learned local dynamics, a bigger model needs to be tested. I doubt there is any model applied in practice with only 8 neurons. ML models applied in practice have thousands to millions of parameters. In order to support this claim, it has to be tested on large models in a systematic way, which is however, challenging to achieve.

As already mentioned above, we extended our analysis to a wider range of architectures. Also, our main architecture is now a bigger network than in the old manuscript (2 layers with 128 neurons each), and we discuss in the conclusion section that other methods might give different results.

Typos

Page 1, line 18, typo "... the widely studied ..."2.

fixed

Page 2, line 5, typo "... the widely In in this paper ..."3.

fixed

Page 4, line 24, typo "The neural networks are trained..."4.

Unfortunately, we were not able to find the typo.

Page 6, line 19, In order to avoid misconception, the following reformulation would help the reader: "Figure 2 shows the training data and the forecast error for a network ..."

We changed to "Figure 1 a,b shows the training data and the attractor reconstructed by the neural network."

5. Page 15 line 3, typo "... of how to test test the reconstruction at training time ... "C7

We deleted the sentence in the new manuscript

**Reviewer #2**

My overall opinion is that whilst the questions the paper addresses are interesting and important, and the results are well-presented on the whole, the experiments performed are not very close to how neural networks would be applied in reality, so it's not clear if they have real-world applications (even notwithstanding the simplicity of the systems being studied). In particular, the performance of the neural networks on reproducing the training data often looks so poor that they would not be used in an application, or the training performance is not presented in enough detail, so it's not clear that the results would apply to real-world applications that would require good validation performance. Also, the changes in forcings applied in the second part seem a lot larger than for applications that neural networks might be considered for. In addition, the authors' experiments on the Lorenz '63 system have used a particular neural network design, where the full state is predicted at every time step (it's not clear if this is the case for the Lorenz '95 system as well) – this may be expected to work worse than other designs where only the change in the state is predicted at each step, or where bias-correction of an approximate dynamical model is performed, and the results here are not clearly generalisable to those set ups.

I have given more detailed comments below. I think the work could eventually be publishable, if the comments are adequately addressed. Since my comments are quite substantial, it could be acceptable to just include the L63 experiments on training on part of the attractor and the L95 experiments on response to forcing – the L63 experiments on responding to a parameter change seem less applicable to real-world cases like predicting climate change. I do encourage the authors to continue with this line of investigation, which I think is potentially very valuable.

We thank you for your positive outlook on our study. We have decided to still include a "forcing" experiment with the L63 system. Even though we agree that it is less applicable to real-world cases, we think it is interesting to include it so as to broaden the range of examples we provide in the study. The study now addresses both questions (extrapolation to left-out phase space regions and learning of external forcing) on both systems, which we think is the most logical and complete approach

**Most significant comments:**

**1. Training performance of neural networks:**

a. For the Lorenz '63 experiments, the ability of the trained neural network to reproduce the attractor appears quite poor (fig.1), and much worse than in the results presented by Zhang (2017), whose work the authors say they are following. A model with such performance would not be used in any real-world application I can think of, and the later results may be much worse than for a well-trained system. We thank you for your positive outlook on our study. We have decided to still include a "forcing" experiment with the L63 system. Even though we agree that it is less applicable to real-world cases, we think it is interesting to include it so as to broaden the range of examples we provide in the study. The study now addresses both questions (extrapolation to left-out phase space regions and learning of external forcing) on both systems, which we think is the most logical and complete approach

It should probably be checked that all of the important steps in the prior work were followed, and if that does not resolve the problem, different architectures tried (e.g. using more neurons) until a good simulation of the attractor is produced.

In the new manuscript we have abandoned the architecture from Zhang (2017), and instead used an objective tuning procedure (similar to the one we already used for the L95, see Appendix A in the new

manuscript). The network that came out of this tuning procedure produces a much better reconstruction of the L63 attractor (fig.1 b)

b. More diagnostics for the performance of the Lorenz '95 networks on the training data and the equivalent portion of the test data should be presented to indicate the system's performance and the degree of overfitting e.g. MAE of single-timestep predictions relative to the variance of the system's tendencies.

We now included a plot that shows the distribution of MAE of 1 timesteps predictions, both on the training and on the test data. For reference, we also included the distribution of the mean absolute tendency of the model. (fig. B1c). As can be seen in the figure, the errors are much smaller than the tendencies, and there is only very slight overfitting.

c. In the experiments testing how well the networks capture the response to a changing forcing, it needs to be shown how well the networks reproduce the trend in the training data. For the results to be applicable to predicting the path of global warming, for example, there should be a discernible trend in the training data and the neural networks should reproduce it with an accuracy similar to what climate models achieve, else they would be deemed to be unsuitable for use in prediction.

Our experiments were not designed to allow the network to predict any type of trend. The idea is to test whether the networks can learn the influence of an external forcing (which might have a trend) on the short-term dynamics. Indeed, in climate change, changes in variability are arguably as important as long-term trends, even for temperature (e.g. https://journals.ametsoc.org/doi/full/10.1175/JCLI-D-18-0462.1).

**2. Forcing experiments:**

a.

• The changes in the forcing terms are rather large compared to the effects expected from anthropogenic climate forcing, for example, and I'm not aware of another real-world case where neural networks would be considered for modelling the effects of such large changes in forcing, so it's not clear to me that these results have real-world applicability. For context, anthropogenic radiative forcing of the climate system is projected to be up to a few percent of solar radiative forcing, and in the scenarios with the largest climate changes, total global warming is around 5x what has been seen in the 20 th century, and comparable in size (though not rate) to changes in between ice ages (so we arguably have some data for testing whether models can simulate such large changes well). In the Lorenz '63 experiments here, the change in the sigma parameter (meant to be analogous to radiative forcing of climate?) changes by a factor of 2. In the Lorenz '95 experiments, the forcing change is enough to change the system from being periodic to highly turbulent, which is a much larger

qualitative change than expected from climate change. I wouldn't have expected neural networks to perform well at the tasks set, namely simulating systems that are very different from what they've been trained on, so these results don't seem to provide much new information.

We agree that our original experiments were rather extreme. We have now completely redesigned the forcing experiments. Instead of relatively arbitrarily picking out a "high" and a "low" forcing, we now trained the networks on 6 different training regimes (3 relatively narrow, and 3 broader forcing regimes). For each training regime, the network is then tested on a wide range of new regimes. (see fig 7 and 8 in the new manuscript). We believe that these new experiments provide much better insights in the abilities (and inabilities) of the networks to learn external forcings.

• It seems reasonable to think that neural networks could perform better at simulating the effects of smaller forcing changes, that are more comparable to those in real situations. It would be interesting to test whether the neural networks can reproduce the effects of forcing at the level seen at the end of their training period (relevant for attributing observed weather events to climate change, for example e.g. National Academy of Sciences, 2016, "Attribution of Extreme Weather Events in the Context of Climate Change") and if so, how far beyond the range of forcing they were trained on can they make good predictions for? (c.f. the Paris climate agreement global warming targets of 1.5C and 2C, which are ~1.5x and ~2x the observed warming – it would be interesting to know if neural networks could provide results that are at all useful for predicting the effects of forcing changes of that magnitude.)

As mentioned in our comment above, our forcing experiments have been completely redesigned. These experiments now address similar questions as you proposed (in the scope of the simplified systems we are using).

• As a further comment, it doesn't seem likely that neural networks could learn the effects of forcings outside the range of the training data without having additional information about the effects of larger forcings e.g. the radiative effects of CO2 in the climate change context. So it seems a priori likely that for the given setup the

performance will deteriorate as the forcing becomes larger. Perhaps the experiments here could demonstrate this, but I don't think it would be that surprising.

We understand your point, but in a purely data-driven approach, this is exactly what one would want to do, e.g. to extrapolate the influence of CO2 forcing to higher values outside the training regime. We don't think it is possible to a-priori answer this question. As our new forcing experiments show, the networks are to some extend able to extrapolate the influence of large forcings, albeit only when trained on wide ranges of forcing.

b. The finding that including information about the forcing as an input often worsens performance seems surprising. One reason could be that the network architecture was tuned to optimise performance without the forcing input, and a larger architecture may be needed to perform well with this information. To be a fair test, the network architecture search should be repeated for the networks using forcing as an input – this may be especially relevant for the L63 case, where the network used is quite small. The statement in the discussion that "it may be better not to include the forcing variable as network input" does not seem well-justified due to this, and also because I do not see how in principle a neural network could predict the effect of a change in forcing if it is not given information about the forcing.

We have now repeated the network architecture search for the L95 model also using the forcing as input. Also, due to our new forcing experiments, we removed the statement "it may be better not to include the forcing variable as network input".

Finally, in our new setup, we use the networks not using the forcing information as input only as baseline, as indeed is not expected that they could learn the influence of any forcing.

3. It should be made clear in the abstract and conclusions that the results apply for a particular choice of neural network design, namely feedforward networks predicting the system state at time t+1 given the state at time t (it's not clear if this is also the case for the L95 experiments, and this should be clearly stated). Also, the L63 experiments testing whether neural networks could represent the system in one wing of the attractor having been trained on the other wing only used sigmoid activation functions.

We completely agree that our results do only apply for feed-forward networks. We now mention this both in the abstract and the discussion sections. , and we also changed the manuscript title accordingly. We indeed used a full-state prediction also for the L95. This is now mentioned in the manuscript. The new architecture we chose for the L63 model uses ReLU activation functions. We repeated some of the experiments also with networks that forecast tendencies instead of full states. This leads to similar results (see Appendix C1).

• Predicting the whole state at every time step may be expected to work worse than other designs where only the change in the state is predicted at each step (e.g. Dueben and Bauer, 2018), or where bias-correction of an approximate dynamical model is performed (Watson, 2019, https://doi.org/10.1029/2018MS001597). This is because in these cases, a lot of the variance in the quantity being predicted is removed, so minimising the RMSE in training may work better to give a system that is capturing the important aspects of the variability. These different methods should also be discussed, and the abstract and conclusions should say that the results may not apply to methods like these.

We now mention this in the discussion section. However, as already mentioned, we repeated part of the experiments with networks predicting tendencies, and these were comparable to the ones predicting full states.

• The choice of sigmoid activation functions for the L63 network may be relevant for the result that the network will not make predictions outside of the range of its training data because sigmoids saturate, and may have trained to saturate at prediction values that are not far outside the boundaries of the training data region, making it difficult for the neural network to predict values outside this region. It would be good to check what happens when using an activation function that does not saturate e.g. ReLu. (Though I still

wouldn't expect it to work well when so much data is left out from training – but I'd still find the result interesting, particularly if done with a system that predicted the tendency rather than the whole state).

Our new architecture uses ReLu functions. Also, as already mentioned, we repeated part of the experiments with networks predicting tendency.

Other comments

1. p.1 L19 It would seem relevant to include citations to other recent studies using neural networks to simulate the Lorenz 95/6 system (Chattopadhyay et al., 2019, https://doi.org/10.31223/osf.io/fbxns; Watson, 2019, https://doi.org/10.1029/2018MS001597).

We now mention these 2 studies in the introduction, and we also refer to the latter in the discussion section.

2. p.2 L10 - Perhaps also mention Lorenz '95 is sometimes called Lorenz '96 we now mention this in section 3.1

3. p.2 L10-12 – Some context here may be useful. For paleoclimate variability and the oceans over multiple decades, yes, but it's less likely to be the case for the atmosphere that unforced variability would be far outside what we've observed.

we now added "Our knowledge of the high-frequency evolution of the climate system issues from comparatively short timeseries, which only explore a small subset of the possible states of the system. This is particularly true for the ocean, which has much longer characteristic timescales than the atmosphere, and for applications to paleoclimatic variability." in the introduction in order to give more context (p.2 L20).

4. p.3 L9-10 Training where no large regions of phase space is left out seems to be the most realistic case for atmospheric modelling, which is what is referred to. The experiments may be relevant for e.g. ocean modelling, where time scales are much longer. (I do think they are inherently interesting, as well.)

Thanks for pointing out that in ocean modelling the problem of not having all phase space covered in training is a realistic case. We now added " In a climate science context, this would for example be relevant for the ocean. The latter's long characteristics timescales imply that observational datasets may cover only part of the phase-space. It is also relevant in forecasting extreme events in the atmosphere." in the introduction. (p.4 L7).

5. p.3 L25 I'm not sure if all readers would be familiar with the Lorenz butterfly - perhaps refer to a figure.

We now refer to fig. 1a which shows a long L63 integration

6. p.3 L26 A better description of the solver is needed e.g. what software package is this from? Reference?

We used the implementation provided by scipy. We now refer both to scipy and to the original paper describing ODEPACK.

7. p.4 L1 Lorenz95 seems to be more often used to describe the 2-level model Lorenz introduced in the same paper. Perhaps use a different name to make it clear you are considering the 1-level version. (As an aside, the 2-level model could be used to test how well neural networks perform compared to a "truncated" model of the system i.e. the 1-level model – this would address whether neural networks can improve upon other reasonable

dynamical models, which is a key question.)

We now mention that the Lorenz95 model is sometimes referred to as Lorenz96, and also explicitly mention that we use the simple version of the Lorenz95 model (without additional levels).

Also, we added "Additionally, it would be interesting to extend the analysis to the 2-level version of the Lorenz95 model, which would allow to also compare the networks to ``truncated''' in the conclusion section. (p.20 L33).

8. p.4 L4 What is the behaviour of the system like with N=40? A plot of a time series or similar may be helpful.

An example of a Lorenz95 model integration is shown in fig. B1 c (left panel). We now explicitly refer to this figure in section 3.1

9. p.4 L8-12 More detail seems necessary to make the results reproducible - e.g. Were input variables normalised? Regularisation? Minibatch size? Learning rate? Stopping rule?

All input variables were normalized to zero mean and unit variance. We used no regularization techniques except for early stopping (after the validation loss has not increased for 4 epochs). Mini-batch size is 32. This is now all mentioned in the methods section and the appendix.

10. p.4 L21-22 Why not give F as one value in the L95 system? (Analogous to using CO2 concentration as an input to a climate model.)

This is not possible in a purely convolutional architecture such as we use. It is standard practice to provide inputs that do not have a spatial pattern (such as our forcing) in the way we do, namely extending it to all points and adding it as an additional layer. For example, it is used to represent the binary game state in googles AlphaGo Zero (https://www.nature.com/articles/nature24270.pdf), and to feed the day of the year in addition to 3d fields in https://www.geosci-model-dev.net/12/2797/2019/

11. p.5 L8 – how big are the grid boxes being used?

They have a size of 0.3x0.3x0.3 on the normalized data. We now mention this in section 3.4

12. p.5 L8-9 – what does "normalized data" mean here?

We added "(normalization based on the training set, the output of the networks is always in the normalized domain)." to make this clearer. (p.6 L10)

13. p.5 L22 For testing 1), the trajectory will tend to the fixed point and only reach it after infinite time, so it's not clear that it could be identified by looking for two points to be exactly equal - perhaps set a threshold on how small a tendency is acceptable and check if the tendency magnitude falls below this and does not rise above it again?

In the new manuscript, we do not use this type of selection any longer.

14. p.5 L23-24 What is the motivation behind 2? To identify periodic behaviour? But couldn't points on a periodic orbit fall at slightly different points on that orbit and so avoid detection?

In the new manuscript, we do not use this type of selection any longer

15. p.6 L1-2 The information on training here could perhaps be merged with the earlier section about training.

We prefer to leave it in this section, as we believe it follows well the logical flow of the paragraph.

16. p.6 L5 what does "consecutive forecasts" mean as opposed to just "forecasts"?

With this we want to point out that we use one forecast to initialize the next one. We now added "(via feeding the forecast back to the input)" to make this clearer. (p.6 L28)

17. p.6 L5-6 See earlier comment about the simulated attractor not looking reasonable.

Due to our new main network architecture, the simulated attractor now closely resembles the true one (fig. 1b).

18. p.6 L9 How big are these errors compared to an average tendency magnitude? Also, could errors on the wings be larger because tendencies are typically larger? Perhaps showing a relative error would be more useful.

We now included a plot of the typical tendencies in (fig. 1d) and refer to it in the text ("To put forecast errors throughout the paper in context, panel d) shows the tendency (change between successive timesteps) of the model in different regions of phase space.") (p7 L3).

19. p.6 L14-15 I didn't get the meaning of "the points...included point".

If we simply remove points from the Lorenz63 model run, and then split our training set into inputs and targets (via shifting by 1 timestep), then there would be input-target pairs were the model "jumps", because a piece of the trajectory in-between had been cut out. We added "(this is necessary because we removed parts of the models' trajectory)" to clarify this. (p8 L1).

20. p.8 L11-12 The errors look quite a bit smaller to me, particularly for the case of fig.3f.

We discuss this in more detail in the new manuscript, since with our new network architecture it is possible to switch off more neurons at once, and the results are more complicated.

21. p.8 L12-14 I don't feel convinced by this analysis. Even if the neural network had learnt to fit the whole attractor well, there may be some neurons whose activations only vary in part of the phase space due to the values of the inputs, and fixing their values would of course be expected to degrade forecasts in areas of the phase space where they do matter.

We think this might have been a misunderstanding. In this analysis, we inspect the network that was trained on the whole attractor, and that also works well on the whole attractor. So we actually know that the network has learned the whole attractor. To make clearer that we analyzed the network trained on the whole attractor, we added *"For this, we inspect the network that was trained on the whole attractor (and forecasts well on the whole attractor)."* in section 4.1.2 (p8 L20).

22. p.11 L5-6 This experiment is not really like rising anthropogenic climate forcing because the sign of the "forcing" depends on whether y is larger than x. An additive forcing term like that used for the L95 experiments and by Palmer (1999, "A nonlinear dynamical perspective on climate prediction", J. Clim.) would be a better analogy.

We removed the corresponding sentence. Additionally, we now discuss in the conclusion section that our experiments are only very remotely related to climate forcings.

23. p.11 L6-7 Changing the number of time steps looks to change the rate of change in the forcing as well. This is something that should be made clear. The results will conflate the effects of changing the amount of training data and the rate of change of the forcing. Perhaps a test could be done for the case with a lower rate of change of forcing with the lower number of time steps used in training to see the effect of changing each aspect of the set up.

In our new forcing experiments, we use a fixed number of training timesteps, so this is not an issue anymore.

24. p.11 L17 It's not clear to me if different training and testing runs are produced for each experimental repeat.

No, only the training is repeated (with exactly the same datasets). We now explicitly mention this.

25. p.13 L2 Given that the performance of the Lorenz systems with fixed F also vary between the situations with different training data lengths, could the different rates of forcing change in each case be affecting the system's behaviour in some important way?

In the new manuscript, we use only the long training length, so this is not an issue anymore. In the old manuscript it might have had an influence, but since the network was trained on short-term forecasts, even for the shorter training sets the individual samples can be seen as having nearly fixed F, because the forcing varied over a long time period only.

26. p.15 L25 - "layers" should be "neurons"?

Yes indeed, thanks for pointing this out. We changed "layers" to "neurons".

27. fig.2 - It should be pointed out clearly that the colour scales are different.

This is now pointed out in the caption

28. fig.3 - It's a bit unclear to compare the distributions across panels b and e to see which neurons change behaviour more - it might be better to put the values side by side on the same axes.

Thanks for this suggestion, we have now put the distributions for both wings on the same axis, with different colors.

29. fig.3 – panel references are wrong in the caption. It also needs to be clearer about what the distributions in b and e are.

We changed the figure to accommodate the distribution plots for the now larger network. Also, we now explain the distributions better in the caption ("*boxplots showing the distribution of neuron activations per neuron* ....")

30. fig.5 - It needs to be made clear that single-timestep prediction errors are being shown. It would be more interesting to see metrics of longer-range forecast skill and how well theattractor is simulated, since in the end single-timestep predictions are not the target and may not indicate that well the performance on longer time scales.

31. fig.5a - I suggest using a different colour for the vertical lines.

32. fig.5 - It would be better to write "Ntrain" in exponential notation.

33. fig.5 - The orders of the Lorenz systems could be reversed so they go start to end, the same as for the neural networks.

34. fig.5 - There are no bars on the simulations with the Lorenz system, so have you used the same Truth runs for each experiment, so you could only run the Lorenz systems once, or are the bars just not visible? It would be good to make small bars visible by giving their ends a width so they stick out.

Answer to all comments above regarding fig.5: We do not use the former fig.5 anymore. We believe that the figures for the new forcing experiments are clearer, and they also include two metrics of attractor reconstruction (mean and standard deviation).

35. References – some references have repeated DOIs.

Fixed

**Reviewer #3**

**General comments:**

A first remark is that the first experiment seems quite artificial. The Lorenz 63 model is an extremely idealized model with a very peculiar bifurcation structure and distinct symmetries, making it less than ideal to represent a realistic general circulation model. A more realistic model which also displays regimes such as that of Charney and Straus (1980), would probably have been more suitable for this particular experiment. Of course, the authors mention these caveats, but in the end there are so many caveats that it seems that no conclusion can be drawn at all related to realistic climate models. Moreover, it is not because some insights related to machine learning map to more complex models, that this is a universal property. I'm not sure how the authors can address this issue without performing the analysis for a more realistic model. Nevertheless, I appreciate the value of this experiment, albeit in a more theoretical context of dynamical systems theory.

We completely agree that the Lorenz63 model is a highly idealized system. We have chosen it due to its widespread use in the study of chaotic dynamical systems, and the fact that it is very easy to visualize and inspect. However, we have now also extended our first experiment to the Lorenz95 system. While this system is also highly idealized, one may argue that it is more closely related to geophysical systems. We agree that it would be interesting to extend the analyses to more complex systems such as the Charney and Straus model, or to simple GCMs. However, considering the difficulty (not at least the required computation time), we think that this is not in the scope of the current study, which we designed for the Lorenz systems. We believe that even when focusing only on the Lorenz systems, this study provides valuable information for the use in more complex systems.

Secondly, the result of the forcing experiment of the Lorenz 95 model seems hardly surprising, as the forcing parameter is varied from the periodic regime to the turbulent regime. One cannot expect a neural network to predict qualitatively completely different behaviour.

We agree that our original experiments were rather extreme. We have now completely redesigned the forcing experiments. Instead of relatively arbitrarily picking out a "high" and a "low" forcing, we now trained the networks on 6 different training regimes (3 relatively narrow, and 3 broader forcing regimes). For each training regime, the network is then tested on a wide range of new regimes. (see fig 7 and 8 in the new manuscript). We believe that these new experiments provide much better insights in the abilities (and inabilities) of the networks to learn external forcings.

Finally, it seems to me that similar studies must have been performed in the literature on neural networks, though not necessarily in the context of geophysics. I would encourage the authors to explore the literature on this. The recent groundbreaking success of deep learning was only possible thanks to the move from few to many hidden layers, and it appears that large deep learning networks have better generalization properties

than smaller ones. It would therefore also be interesting to repeat the exercise for a deep neural network. See for example the work by Novak et al. (2018) or Wu et al. (2017) who investigate the source of these generalization properties, and references therein.

We have added a new section in the introduction ("1.2 Related work on generalization properties of neural networks") that gives a better overview of the general literature on generalization properties of neural networks. We have further tested several architectures, including ones with wider layers, in our experiments.

Specific comments and typographical errors:

p.1, L 9-10: In the abstract, the authors conclude that "These results outline challenges for a variety of machine-learning applications. [...]". The word "outline" (in the sense of summarize) goes a bit too far since the results shown are for two highly specific models and a very artificial set-up (an entire wing missing, training in periodic regime). I would just say that the results provide some examples.

We changed it to "point to potential limitations"

Figure 3: Labels in the caption don't match with the relevant subfigures.

fixed

p. 11, L 16: lorenz -> Lorenz (2x)

fixed

p. 15, L 1: However, also the alternative methods suffer -> However, the alternative methods also suffer

We changed the paragraph to reflect the new results, and the sentence does not appear anymore.

p. 15, L 3: test test -> test

We changed the paragraph to reflect the new results, and the sentence does not appear anymore.

Check spelling consistency: throughout the manuscript, generalize / generalise are both used

**we changed all to generalize**

[revised manuscript text omitted]